# LASE: Learned Adjacency Spectral Embeddings

**Sofía Pérez Casulo**                                                    *sperez@fing.edu.uy*
*Facultad de Ingeniería*
*Universidad de la República*

**Marcelo Fiori**                                                         *mfiori@fing.edu.uy*
*Facultad de Ingeniería*
*Universidad de la República*

**Federico Larroca**                                                      *flarroca@fing.edu.uy*
*Facultad de Ingeniería*
*Universidad de la República*

**Gonzalo Mateos**                                                  *gmateosb@ece.rochester.edu*
*Department of Electrical and Computer Engineering*
*University of Rochester*

**Reviewed on OpenReview:** *https://openreview.net/forum?id=J65NBLWrmh*

## Abstract

We put forth a principled design of a neural architecture to learn nodal Adjacency Spectral Embeddings (ASE) from graph inputs. By bringing to bear the gradient descent (GD) method and leveraging the technique of algorithm unrolling, we truncate and re-interpret each GD iteration as a layer in a graph neural network (GNN) that is trained to approximate the ASE. Accordingly, we call the resulting embeddings and our parametric model Learned ASE (LASE), which is interpretable, parameter efficient, robust to inputs with unobserved edges, and offers controllable complexity during inference. LASE layers combine Graph Convolutional Network (GCN) and fully-connected Graph Attention Network (GAT) modules, which is intuitively pleasing since GCN-based local aggregations alone are insufficient to express the sought graph eigenvectors. We propose several refinements to the unrolled LASE architecture (such as sparse attention in the GAT module and decoupled layerwise parameters) that offer favorable approximation error versus computation tradeoffs; even outperforming heavily-optimized eigendecomposition routines from scientific computing libraries. Because LASE is a differentiable function with respect to its parameters as well as its graph input, we can seamlessly integrate it as a trainable module within a larger (semi-)supervised graph representation learning pipeline. The resulting end-to-end system effectively learns "discriminative ASEs" that exhibit competitive performance in supervised link prediction and node classification tasks, outperforming a GNN even when the latter is endowed with open loop, meaning task-agnostic, precomputed spectral positional encodings.

## 1 Introduction

Graphs are natural models of relational data, spanning domains as diverse as social networks, molecular biology, recommender systems, and knowledge graphs. Graph representation learning (GRL) has emerged as a powerful framework for extracting meaningful patterns from such structured data, enabling downstream machine learning tasks. At its core, this approach seeks to encode nodes (also edges, or entire subgraphs) into low-dimensional vectors –known as *node embeddings*– that capture both local and global graph structural properties. These learned representations facilitate scalable, flexible, and effective learning, driving advancements in applications such as drug discovery, fraud detection, and personalized recommendations.

**Objectives, context, and motivating challenges.** Here we revisit *spectral* node embeddings derived from the eigendecomposition of the graph's adjacency matrix; see e.g., (Lim et al., 2023) and note extensions to Laplacian embeddings are straightforward. Spectral embeddings are central to unsupervised node clustering methods (von Luxburg, 2007), which intuitively motivates why information encoded in the graph eigenvectors may be used to specify vertex positions in latent space. This intuition is further justified if we assume that the graph's structure stems from a Random Dot Product Graph (RDPG) (Athreya et al., 2017), a generative model that subsumes the classic Stochastic Block Model (SBM) and which has close ties to the general class of latent position network models (Hoff et al., 2002). RDPGs associate a vector $\mathbf{x}_i \in \mathcal{X} \subset \mathbb{R}^d$ to each node, and specify that an edge exists between nodes $i$ and $j$ with probability given by the inner-product $\mathbf{x}_i^\top \mathbf{x}_j$ of the corresponding embeddings; independently of all other edges. That is, the random adjacency matrix $\mathbf{A} \in \{0,1\}^{N \times N}$ has entries $A_{ij} \sim \text{Bernoulli}(\mathbf{x}_i^\top \mathbf{x}_j)$, where $N$ is the number of vertices. Unsurprisingly, the legacy solution to the embedding problem of estimating nodal positions from an observed graph is given in terms of the adjacency matrix eigenvectors (Athreya et al., 2017); see Section 2.1. Note however that eigendecomposition of the adjacency matrix is not feasible when edge data are partially missing (e.g., in link prediction tasks). Furthermore, node embeddings need to be recomputed from scratch in inductive settings, which may be a significant computational burden.

Our main contribution is the principled design of a neural architecture to learn these *Adjacency Spectral Embeddings (ASE)* solely from (undirected and unweighted) graph inputs, and which overcomes the aforementioned shortcomings of eigendecomposition-based spectral embeddings. As we discuss in Section 5, there have been previous related efforts to learn so-termed nodal Positional Encodings (PEs), with the distinct goal of enhancing the expressive power of GNNs. However, virtually all these works *precompute* a set of PEs which are then fed as inputs to a neural architecture that, for instance, decouples how to refine the PEs and the node representations in a GNN (Dwivedi et al., 2022); or, (in a way akin to an autoencoder) train a GNN followed by a multilayer perceptron to reconstruct these input PEs (Cantürk et al., 2024). Again, this pre-computation step may prove too costly for large graphs, and needs to be carried out repeatedly in inductive settings that embed multiple graphs from some underlying distribution. Our goal is to avoid this overhead altogether and design a model that learns ASEs in an unsupervised manner. Post training, embeddings of new graphs are efficiently obtained via a forward pass through a neural network (NN).

Another challenge to this end lies in the impossibility of computing the eigendecomposition of $\mathbf{A}$ (i.e., the ASE) through a GCN, at least in several important settings. For instance, consider a symmetric SBM graph with two equally-sized communities that have the same connection probabilities. Each node will locally "see" the same structure, and if the input node features have the same characteristics across clusters, the GCN will fail to generate distinct node embeddings for vertices that belong in different communities. This pitfall can be formalized by using graphons, a fairly general non-parametric model for large graphs. In particular, (Magner et al., 2020) identified the degree profile of the graph (i.e., the expected degree of each node) as the feature that enables a GNN to distinguish between two graphons. The Graphon Neural Network framework is particularly illuminating (Ruiz et al., 2020), and these expressive power conclusions still hold true for more general kernel-based graph generative models (Keriven & Vaiter, 2023). See Section 2.2 for an illustrative example and associated discussion about this inherent limitation of GNNs that motivates our work. We thus contend that a new architecture is needed when it comes to learning to approximate ASEs. Our objective here is not to propose a new state-of-the-art PE, but rather to broaden the applicability of ASE –whose merits for statistical inference have been well documented; see e.g., (Athreya et al., 2017).

**Technical approach.** Adopting a low-rank adjacency matrix factorization perspective to the spectral embedding problem, one can bring to bear the gradient descent (GD) method that is provably locally convergent to the ASE solution (Fiori et al., 2024). The optimization formulation is flexible to accommodate embeddings of partially observed graphs, and the GD solver is competitive in terms of computation cost when benchmarked against standard eigendecomposition libraries. Starting from these GD iterations and leveraging the technique of algorithm unrolling (a.k.a. deep unfolding) (Gregor & LeCun, 2010; Monga et al., 2021), we truncate and re-interpret each iteration of the algorithm as a layer in a (graph) NN that can be trained to approximate the ASE. Accordingly, we call the resulting embeddings and our parametric model Learned ASE (LASE), which is interpretable, parameter efficient, robust to inputs with missing edges, and offers controllable complexity during inference.

With regards to interpretability, LASE layers entail a superposition of GCN and *fully-connected* Graph Attention Network (GAT) modules (Veličković et al., 2018; Shi et al., 2021), which is intuitively pleasing since (as argued before) GCN-based local aggregations alone are insufficient to express the sought graph eigenvectors. Furthermore, LASE inherits desirable features of its GD blueprint, including the ability to operate in streaming scenarios where nodal representations must be continuously tracked (Fiori et al., 2024), as well as when substantial edge information is missing due to e.g., sampling, memory, or privacy constraints. We empirically show that refinements to the vanilla LASE architecture (such as sparse attention in the GAT module) can result in favorable approximation error versus computation tradeoffs –inference times are over an order of magnitude faster than GD, and even outperform heavily-optimized routines to estimate the top eigenvectors in scientific computing libraries (Chung et al., 2019). Accordingly, we find LASE can be particularly useful in scenarios where limited computational resources render eidencomposition of (multiple) moderately large graphs excessively time consuming, or, outright infeasible depending on the number of considered embedding dimensions and edge sparsity levels. Finally, since the LASE architecture combines GCN and GAT modules, it can be naturally instantiated on different graphs. We exploit this flexibility to reduce computation costs by training LASE on smaller subgraphs sampled from the larger one to be embedded. We also empirically show how the learned weights are robust to modest shifts in the underlying graph distribution. All in all, LASE yields node embeddings that are robust to missing data as well as minor distribution shifts, at a fraction of the cost of prior art and without sacrificing approximation accuracy.

On top of unsupervised learning of ASEs that are useful e.g., in graph visualization, clustering, hypothesis testing (Athreya et al., 2017), or change-point detection (Marenco et al., 2022), LASE may also be used in an end-to-end fashion for other supervised tasks such as link prediction of node classification. Indeed, since LASE is a differentiable function with respect to its parameters as well as its graph input, we can seamlessly integrate it as a trainable module within a larger (semi-)supervised GRL pipeline. That is, instead of using pre-trained LASE embeddings along nodal features as inputs to e.g., a GNN, consider instead training an *end-to-end* system whose loss function takes into account both the task at hand (e.g., link prediction) as well as the reconstruction errors $|\mathbf{x}_i^\top \mathbf{x}_j - A_{ij}|$ between the output of the LASE decoder and the given adjacency matrix; see (Chami et al., 2022). In the end-to-end system training entails optimizing LASE and GNN parameters *jointly*. We show the resulting architecture effectively learns "discriminative ASEs" that exhibit competitive performance in link prediction and node classification tasks, outperforming a GNN even when the latter is endowed with precomputed, but task-agnostic, spectral PEs.

**Summary of contributions.** In this work, we contribute the following technical GRL innovations and provide experimental evidence to support our claims:

• We propose LASE (Section 3), an unsupervised NN model that can be trained to approximate spectral embeddings of input graphs, even in scenarios where GNNs fail because of their well-documented limitations to express graph eigenvectors. The permutation-equivariant LASE architecture is designed by unrolling a GD method to compute ASEs, i.e., we truncate and map algorithm iterations to interpretable and parameter-efficient NN layers that combine graph convolution and transformer modules. The unrolling offers an explicit handle on complexity leading to faster (post training) inference times and satisfactory ASE approximation error performance, even when a sizable fraction of input graph data are missing; see the tests in Section 4.1.

• We introduce architectural and methodological refinements with an eye towards scalability in resource-constrained settings (Section 3.3). Leveraging sparse attention in the GAT module of the LASE layer enables embedding larger graphs, where spectral decomposition may prove computationally impractical. Since LASE is inductive, we propose a methodology whereby training is performed on smaller sub-graphs. Subsequent inference on the entire graph becomes feasible because of the simplicity of LASE's layers, and yields vectors with minimal performance degradation relative to ASE, the eigendecomposition gold standard.

• In Section 3.4 we integrate LASE in a semi-supervised GRL pipeline that can be trained in an end-to-end fashion, endowing the learned PEs with a discriminative bias for the task at hand (e.g., node classification or link prediction in our tests). Comprehensive and reproducible experiments on numerous synthetic and real-world datasets demonstrate the resulting architecture outperforms a GNN baseline supplemented with precomputed PEs, especially if there are unknown edges in the input graph; see the tests in Section 4.2.

## 2 Background

Here we briefly review the graph spectral embedding problem associated with RDPGs, introduce GNNs and discuss challenges facing their ability to express graph eigenvectors (hence, spectral node embeddings). We also outline the basic ideas behind the algorithm unrolling principle, which we will leverage in Section 3 to design a NN that can learn to approximate spectral embeddings of graphs adhering to the RDPG model.

### 2.1 Adjacency spectral embedding

Going back to the RDPG model for undirected and unweighted graphs $G(\mathcal{V}, \mathcal{E})$ introduced in Section 1, let us state the associated node embedding problem. To this end, start by stacking all the nodes' latent position vectors $\mathbf{x}_i \in \mathbb{R}^d$, $i \in \mathcal{V}$, in the matrix $\mathbf{X} = [\mathbf{x}_1, \dots, \mathbf{x}_N]^\top \in \mathbb{R}^{N \times d}$. Matrix $\mathbf{X}$ is unknown and we would like to estimate it from observed graph data. The RDPG model ties the unknown latent positions with the observations, thus making the aforementioned estimation (i.e., node embedding) task feasible. Formally, recall that RDPGs specify edge-wise formation probabilities $P_{ij} := \mathbf{x}_i^\top \mathbf{x}_j$ for each unordered pair $(i, j) \in \mathcal{V} \times \mathcal{V}$, which correspond to the entries of the rank-$d$, positive semidefinite (PSD) matrix $\mathbf{P} := \mathbf{X}\mathbf{X}^\top$. Recalling $A_{ij} \sim \text{Bernoulli}(\mathbf{x}_i^\top \mathbf{x}_j)$ for $i \neq j$ and because the diagonal entries in $\mathbf{A}$ are zero (we model graphs with no self loops), then we say $\mathbf{A} \sim \text{RDPG}(\mathbf{X})$[1] and have $\mathbb{E}\left[\mathbf{A} \mid \mathbf{X}\right] = \mathbf{M} \circ \mathbf{P}$, where $\circ$ is the entry-wise or Hadamard product and $\mathbf{M} = \mathbf{1}_N \mathbf{1}_N^\top - \mathbf{I}_N$ is a mask matrix with ones everywhere except in the diagonal where it is zero. Given the observed adjacency matrix $\mathbf{A} \sim \text{RDPG}(\mathbf{X})$ of $G(\mathcal{V}, \mathcal{E})$ and a prescribed embedding dimension $d$, the goal is to estimate $\mathbf{X}$. Typically, $d \ll N$ is obtained using an elbow rule on $\mathbf{A}$'s eigenvalue scree plot (Chung et al., 2019).

Since a maximum likelihood estimator of $\mathbf{X}$ is computationally challenging and may not be unique (Xie & Xu, 2023), a natural alternative is to try least-squares (LS) instead (Scheinerman & Tucker, 2010), namely

$$\hat{\mathbf{X}} \in \underset{\mathbf{X} \in \mathbb{R}^{N \times d}}{\arg\min} \left\| \mathbf{M} \circ (\mathbf{A} - \mathbf{X}\mathbf{X}^\top) \right\|_F^2. \tag{1}$$

In words, $\hat{\mathbf{P}} = \hat{\mathbf{X}}\hat{\mathbf{X}}^\top$ is the best rank-$d$ PSD approximant to the off-diagonal entries of the adjacency matrix $\mathbf{A}$, in the Frobenius-norm sense. The RDPG model is identifiable modulo rotations (i.e., for any orthogonal matrix $\mathbf{T} \in \mathbb{R}^{d \times d}$ we have $\mathbf{X}\mathbf{T}(\mathbf{X}\mathbf{T})^\top = \mathbf{X}\mathbf{X}^\top = \mathbf{P}$), and accordingly the solution of (1) is not unique.

Entrywise multiplication with $\mathbf{M} = \mathbf{1}_N \mathbf{1}_N^\top - \mathbf{I}_N$ effectively discards the residuals corresponding to the diagonal entries of $\mathbf{A}$. If suitably redefined, the binary mask $\mathbf{M}$ may be used for other purposes, such as accounting for unobserved edges if data are missing. For instance, in a recommender system we typically have the rating of each user over only a limited number of items. These missing data may be accounted for in (1) by zeroing out the entries of $\mathbf{M}$ corresponding to vertex pairs for which edge status is unobserved.

Typically the mask $\mathbf{M}$ is ignored (and sometimes non-zero values are iteratively imputed to the diagonal of $\mathbf{A}$ (Scheinerman & Tucker, 2010)), which results in a closed-form solution for $\hat{\mathbf{X}}$. Indeed, if we let $\mathbf{M} = \mathbf{1}_N \mathbf{1}_N^\top$ in (1), we have that $\hat{\mathbf{X}} = \hat{\mathbf{V}}\hat{\mathbf{\Lambda}}^{1/2}$, where $\mathbf{A} = \mathbf{V}\mathbf{\Lambda}\mathbf{V}^\top$ is the eigendecomposition of $\mathbf{A}$, $\hat{\mathbf{\Lambda}} \in \mathbb{R}^{d \times d}$ is a diagonal matrix with the $d$ largest-magnitude eigenvalues of $\mathbf{A}$, and $\hat{\mathbf{V}} \in \mathbb{R}^{N \times d}$ are the associated eigenvectors. This workhorse estimator is known as the Adjacency Spectral Embedding (ASE); see also (Athreya et al., 2017) for consistency and asymptotic normality results, as well as applications of statistical inference with RDPGs.

### 2.2 Graph neural networks vis-a-vis spectral embeddings

**Graph convolutional filters, frequency representation, and GNNs.** A GNN layer entails a graph convolution followed by a pointwise non-linearity. The former can be represented as a polynomial

$$\mathbf{X}_{\text{out}} = \sum_{k=0}^{K} \mathbf{A}^k \mathbf{X}_{\text{in}} \mathbf{H}_k, \tag{2}$$

---

[1]We have treated $\mathbf{X}$ as deterministic. One could also assume that the latent position vectors $\mathbf{x}_i \in \mathbb{R}^d$, $i \in \mathcal{V}$ are random and drawn i.i.d. from a suitable "inner-product" distribution $F$ supported in $\mathcal{X} \subseteq \mathbb{R}^d$; see e.g., (Athreya et al., 2017). The (hierarchical) RDPG model that consists of drawing $\mathbf{X} \sim F$ and then $\mathbf{A} \mid \mathbf{X} \sim \text{RDPG}(\mathbf{X})$ is denoted as $\{\mathbf{A}, \mathbf{X}\} \sim \text{RDPG}(F)$.

where the input graph signal $\mathbf{X}_{\text{in}} \in \mathbb{R}^{N \times F^{in}}$ has $F^{in}$ features per node, the output signal $\mathbf{X}_{\text{out}} \in \mathbb{R}^{N \times F^{out}}$ has $F^{out}$ nodal features, and $\mathbf{H}_k \in \mathbb{R}^{F^{in} \times F^{out}}$ is a matrix of learnable filter coefficients. Note that in (2) we use the adjacency matrix and its powers for simplicity; one could use other graph shift operators such as the Laplacian, or, degree-normalized versions of any of these matrix representations of graph structure. See also Appendix A.1 and the tutorial (Isufi et al., 2024) for additional background on graph convolutional filters. In particular, discussion in (Isufi et al., 2024, Section VIII-A) argues that the polynomial representation (2) that is mainstream in graph signal processing circles subsumes "combination and aggregation" operations in several message-passing GNN models, for specific choices of $K$, $\mathbf{H}_k$, and the graph shift operator.

Graph filters can be equivalently represented in the graph spectral (frequency) domain. To see this, consider $F^{in} = F^{out} = 1$ to simplify exposition and once more let $\mathbf{A} = \mathbf{V} \boldsymbol{\Lambda} \mathbf{V}^\top$, where $\boldsymbol{\Lambda} = \text{diag}(\lambda_1, \ldots, \lambda_N)$ collects the graph eigenvalues and $\mathbf{V} = [\mathbf{v}_1, \ldots, \mathbf{v}_N]$ the respective orthonormal eigenvectors. Defining $\tilde{\mathbf{x}} = \mathbf{V}^\top \mathbf{x}$ as the Graph Fourier Transform (GFT) of a graph signal $\mathbf{x}$, it follows that the GFT of $\mathbf{x}_{\text{out}}$ in (2) is

$$\tilde{\mathbf{x}}_{\text{out}} = \left( \sum_{k=0}^{K} h_k \boldsymbol{\Lambda}^k \right) \tilde{\mathbf{x}}_{\text{in}},$$

which is a pointwise equality since $\boldsymbol{\Lambda}$ is diagonal. Furthermore, we may recover the actual output signal via the inverse GFT $\mathbf{x}_{\text{out}} = \mathbf{V} \tilde{\mathbf{x}}_{\text{out}}$. These results are easier to grasp in terms of inner products and orthonormal signal decompositions. GFT coefficients in $\tilde{\mathbf{x}}_{\text{in}}$ correspond to the inner product between $\mathbf{x}$ and the corresponding eigenvectors – i.e., the graph's frequency modes. To carry out the graph convolution, the $i$-th coordinate $\tilde{x}_{\text{in},i}$ is scaled by the filter's frequency response $\tilde{h}(\lambda_i) := \sum_{k=0}^{K} h_k \lambda_i^k$ at frequency $\lambda_i$, to yield $\tilde{x}_{\text{out},i}$. Finally, each eigenvector $\mathbf{v}_i$ is scaled by $\tilde{x}_{\text{out},i}$ and summing over frequencies yields $\mathbf{x}_{\text{out}} = \sum_{i=1}^{N} \tilde{x}_{\text{out},i} \mathbf{v}_i$. Recapping, signal $\mathbf{x}_{\text{in}}$ is decomposed in terms of the columns of $\mathbf{V}$, each component is then filtered via $\tilde{h}(\lambda_i)$, and the output is a new combination of the columns of $\mathbf{V}$ using these filtered coefficients.

These insights carry over to the multi-channel graph filter (2), since $\mathbf{X}_{\text{out}}$ is a linear combination of several single-channel filters. Furthermore, a GNN layer is simply a pointwise non-linear operation $\sigma(\cdot)$ [e.g., $\sigma(\mathbf{x}) =$ ReLU$(\mathbf{x})$] applied to the output of the graph convolution. Apparently, this frequency domain viewpoint suggests eigenvectors of $\mathbf{A}$ play a crucial role in the expressivity of a GNN, as we now further illustrate; see also the studies in e.g., Kanatsoulis & Ribeiro (2024); Ruiz et al. (2024).

**On GNN's ability to express graph eigenvectors and spectral embeddings.** Consider for instance a regular graph where all nodes have the same degree $r$, implying that $\mathbf{v}_1 = \frac{\mathbf{1}_N}{\sqrt{N}}$ is an eigenvector of $\mathbf{A}$ associated to eigenvalue $\lambda_1 = r$. Suppose we want to perform unsupervised node clustering with a GNN. Even if all nodes have the same degree, they can still belong to different classes and for instance prefer connecting to nodes of the same community; see Example 1 for an SBM where nodes in two different communities have the same expected degree. Since we only know the graph's structure and have no informative nodal attributes we could choose as an input signal, we are free to choose whatever signal we want. For instance, $\mathbf{x}_{\text{in}} = \frac{\mathbf{1}_N}{\sqrt{N}}$ or any other constant signal are widely used in the literature; see e.g., (Morris et al., 2019; Xu et al., 2018). It is clear that the GNN (with one or multiple layers) will output $\mathbf{x}_{\text{out}} = \sigma\left( \tilde{h}(\lambda_1) \frac{\mathbf{1}_N}{\sqrt{N}} \right)$, which is also a constant signal and thus ineffective to accomplish our discriminative goal.

The problem lies precisely in using a constant signal as an input, which happened to be one of the eigenvectors, and thus all other information on the graph's structure is lost because of mutual orthogonality of the frequency modes. A popular alternative to reveal further information by exciting all spectral components is to feed the GNN with zero-mean white noise across nodes (Sato et al., 2021; Abboud et al., 2021; Kanatsoulis & Ribeiro, 2024). For instance, by feeding standard white Gaussian noise to the convolutional layer, $\tilde{\mathbf{x}}_{\text{out}}$ becomes another zero-mean uncorrelated Gaussian vector, where now each entry has variance equal to $\tilde{h}^2(\lambda_i)$. The output signal $\mathbf{x}_{\text{out}}$ in the vertex domain is thus a linear combination of the columns of $\mathbf{V}$, with random weights. Interestingly, it is possible to recover, say, the $j$-th eigenvector $\mathbf{v}_j$ through a graph convolution. For high enough $K$, filter coefficients $h_k$ may be chosen so that $\tilde{h}(\lambda_i) = 0$, $\forall i \neq j$ (i.e., a frequency selective or notch filter in the signal processing lingo). Actually, white noise may be used to compute the ASE in (1) through a graph convolution as we show in the following example. However, the example also highlights how this solution is unsatisfactory, since transferability across graph sizes (from the same model) is not possible.

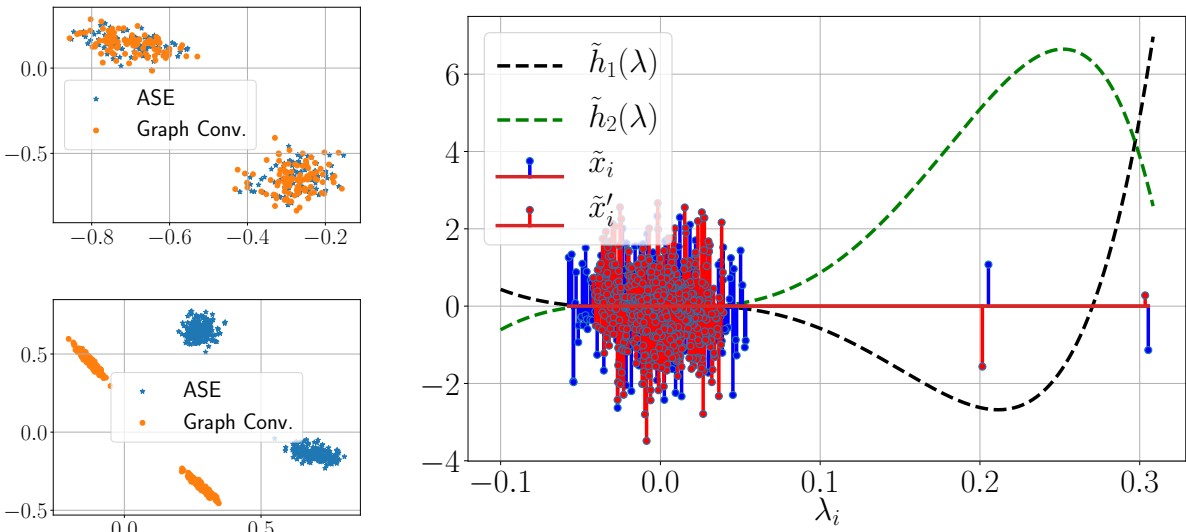

Figure 1: (left) Estimates $\hat{\mathbf{X}}$ in (1) obtained through ASE and a graph convolution trained as in (3). When using a single white noise sample $\mathbf{x}_{\mathrm{in}}$ and graph $G$, the graph convolution-based embedding is quite accurate (top-left). However, the learned filter coefficients cannot be used to estimate the ASE of another graph with more nodes, even if they stem from the same SBM model (bottom-left). (right) Filter frequency responses $\tilde{h}_1(\lambda)$ and $\tilde{h}_2(\lambda)$ for the learned coefficients in (3), as well as the GFT coefficients of both the original signal $\mathbf{x}_{\mathrm{in}}$ and the new sample $\mathbf{x}'_{\mathrm{in}}$. The height of each stem represents the value of the $i$-th GFT coefficient, supported on the corresponding frequency $\lambda_i$. Since both $G$ and $G'$ adhere to the same SBM and we have used a normalized adjacency matrix, their eigenvalues are similar. However, the fact that we were forced to generate new random input samples produced a random combination of the eigenvectors as the output, resulting in inaccurate estimates of $\hat{\mathbf{X}}$ as shown in the bottom-left subplot.

**Example 1 (Symmetric SBM)** Consider a graph $G(\mathcal{V}, \mathcal{E})$ sampled from an SBM with $C = 2$ classes of $N/2 = 100$ nodes each and symmetric inter-class connection probability matrix $\mathbf{\Pi} = \left( \begin{smallmatrix} 0.5 & 0.1 \\ 0.1 & 0.5 \end{smallmatrix} \right)$, where $d = 2$ is thus enough for $\hat{\mathbf{X}}$ in (1). Consider using a GNN to identify to which class each node belongs to. A baseline approach is to compute the ASE $\hat{\mathbf{X}}$ and cluster the resulting vectors – a method that enjoys theoretical guarantees (Athreya et al., 2017). Denote by $\mathbf{\Phi}_G(\mathbf{x}_{\mathrm{in}}; \mathcal{H}) \in \mathbb{R}^{N \times 2}$ the output of the graph convolution using filter coefficients $\mathcal{H} = \{\mathbf{H}_k \in \mathbb{R}^{1 \times 2}, k = 0, \ldots, 20\}$. Given a white noise sample $\mathbf{x}_{\mathrm{in}} \in \mathbb{R}^N$, we want to find filter coefficients $\mathcal{H}^*$ such that [cf. (1)]

$$\mathcal{H}^* = \arg\min_{\mathcal{H}} \|\mathbf{M} \circ (\mathbf{A} - \mathbf{\Phi}_G(\mathbf{x}_{\mathrm{in}}; \mathcal{H}) \mathbf{\Phi}_G^\top(\mathbf{x}_{\mathrm{in}}; \mathcal{H}))\|_F^2, \tag{3}$$

where here $\mathbf{A}$ denotes the adjacency matrix normalized by the number of nodes $N$. The optimal coefficients may be found through backpropagation, which results in two set of filter coefficients (corresponding to each output dimension), and thus in two frequency-domain responses $\tilde{h}_1(\lambda)$ and $\tilde{h}_2(\lambda)$. The resulting $\mathbf{\Phi}_G(\mathbf{x}_{\mathrm{in}}; \mathcal{H}^*)$ superimposed to the ASE solution is depicted in Fig. 1 (top-left). The resulting embeddings are quite similar, with a total reconstruction error equal to 81.7 and 81.6 for the graph convolution and ASE in (1), respectively.

Now suppose that we want to use the learned architecture to embed nodes from another graph $G'(\mathcal{V}', \mathcal{E}')$. Specifically, we consider another SBM graph with the same parameters $\mathbf{\Pi}$, except that now both communities have double the size (i.e., $N'/2 = 200$ nodes each). So even if we wanted to reuse $\mathbf{x}_{\mathrm{in}}$ from the previous setting, we must generate 200 new random samples to populate the input $\mathbf{x}'_{\mathrm{in}} \in \mathbb{R}^{N'}$. The resulting embeddings $\mathbf{\Phi}_{G'}(\mathbf{x}'_{\mathrm{in}}; \mathcal{H}^*)$ are shown in Fig. 1 (bottom-left), while the reconstruction errors are now 202.6 and 164.7 for the graph convolution and ASE, respectively. The reason behind this failure is illustrated in Fig. 1 (right), showing the resulting GFT coefficients of the input signals $\mathbf{x}_{\mathrm{in}}$ and $\mathbf{x}'_{\mathrm{in}}$, supported on the corresponding eigenvalues of $\mathbf{A}$ and $\mathbf{A}'$ respectively. The filter coefficients in $\mathcal{H}^*$, producing $\tilde{h}_1(\lambda)$ and $\tilde{h}_2(\lambda)$, are such

---

**Algorithm 1** An iterative algorithm

---

**Require:** Initialize $\mathbf{X}_0$

   $l \leftarrow 0$

   **while** Convergence criteria not met **do**

      $\mathbf{X}_{l+1} \leftarrow h\left(\mathbf{X}_l, \theta_l\right)$

      $l \leftarrow l + 1$

   **end while**

   **return** $\mathbf{X}_l$

---

Unrolling $\quad \mathbf{X}_0 \longrightarrow h(\cdot; \theta_1) \longrightarrow \cdots \longrightarrow h(\cdot; \theta_l) \longrightarrow \cdots \longrightarrow h(\cdot; \theta_L) \longrightarrow \mathbf{X}_L$

Figure 2: A schematic illustration of the algorithm unrolling principle: given an iterative algorithm (left), each iteration is mapped into a NN layer and the number of iterations is truncated to $L$. The layers are then composed to form a deep NN of depth $L$ (right). Algorithm step-sizes, penalty, or regularization parameters are mapped to NN weights $\{\theta_l\}$, and learned from training data by minimizing a suitable loss function.

that they overfit the original white noise input $\mathbf{x}_{\text{in}}$ in order to produce a good approximation of $\hat{\mathbf{X}}$. Indeed, notice from Fig. 1 (right) that, as expected, these optimal filters end up relying only on the two dominant components $\mathbf{v}_1$ and $\mathbf{v}_2$ (for $\lambda_1 \approx 0.3$ and $\lambda_2 \approx 0.2$), while all other frequency modes are atennuated and effectively discarded. However, when another noise sample is used the input signal's GFT coefficients can change significantly [compare $\tilde{x}_1$ vs $\tilde{x}'_1$ and $\tilde{x}_2$ vs $\tilde{x}'_2$ in Fig. 1 (right)], and hence the resulting outputs will be (different) random linear combinations of the corresponding eigenvectors.

Apparently, the preceding example showcases the sensitivity of the predicted spectral embeddings to the input signal $\mathbf{x}_{\text{in}}$ fed to the convolutional model. One could attempt to train the architecture with several noise samples as input to improve generalization. However, the problem remains, since for any given set of parameters $\mathcal{H}$, the output will still be a random linear combination of the eigenvectors and thus generalization is not possible. An alternative is to instead consider a (non-linear) statistic of this random output. For instance, the expected value of the entry-wise square of the output of a convolutional layer; i.e., $\mathbb{E}\left[\mathbf{\Phi}_G^2(\mathbf{x}_{\text{in}}; \mathcal{H})\right]$ (Kanatsoulis & Ribeiro, 2024). However, in this symmetric case we obtain a constant signal (see Appendix A.2 for details), which takes us back to the discussion at the opening of this section. Since the node embedding (i.e., matrix factorization) problem (1) can be solved via factored GD and exhibits robust convergence properties even from random initializations (Fiori et al., 2024; Chi et al., 2019), our idea is to mimic those iterations in a custom NN obtained via algorithm unrolling – the subject dealt with next.

## 2.3 Algorithm unrolling

The algorithm unrolling (or deep unfolding) technique was pioneered by (Gregor & LeCun, 2010) and consists of truncating and mapping an iterative optimization algorithm into a NN architecture. Each iteration of the algorithm becomes a layer of the NN, and subsequently composed to form a deep NN as depicted in Fig. 2. The mapping process was quite natural in the sparse coding context of (Gregor & LeCun, 2010), since the workhorse iterative shrinkage-thresholding algorithm (ISTA, Beck & Teboulle (2009)) boils down to repeated: (i) linear updates stemming from the gradient of the quadratic LS loss; followed by (ii) a pointwise nonlinear (ReLU-like) soft-thresholding operator (the proximal operator associated to the $\ell_1$-norm regularizer). A forward pass through an unrolled NN emulates the execution of the iterative algorithm for a finite number of iterations. Optimization algorithm step-sizes, penalty, or regularization parameters are mapped to NN weights, and therefore learned from data through end-to-end training using backpropagation.

One can view this process as effectively truncating the iterations of an asymptotically convergent procedure, to yield a template architecture that learns to approximate solutions of optimization problems with substantial computational savings relative to the optimization algorithm. For instance, results in (Gregor & LeCun, 2010) show that the learned version of ISTA outputs sparse image representations of comparable performance, but 20x faster than the iterative solver. Moreover, as discussed in (Monga et al., 2021) the unrolled network is amenable to architectural modifications, such as letting the learned parameters vary across different layers instead of being shared. This deviation from the original iterative algorithm can result in major performance gains, as shown in various applications; see e.g., (Ye & Mateos, 2022; Pu et al., 2021;

Wasserman et al., 2023; Monga et al., 2021). Finally, the strong inductive biases encoded in the unrolled NN makes these models both parameter and sample efficient (Monga et al., 2021).

# 3 Learned Adjacency Spectral Embedding (LASE)

## 3.1 Problem statement

Suppose we are given training samples of adjacency matrices $\mathcal{T} := \{\mathbf{A}^{(i)}\}_{i=1}^{T}$ adhering to an RDPG($\mathbf{X}$) model. These adjacency matrices may be partially observed, in which chase each $\mathbf{A}^{(i)}$ is accompanied by a mask $\mathbf{M}^{(i)} \in \{1,0\}^{N \times N}$ encoding which entries $A_{ij}^{(i)}$ are missing; see Section 2.1. Let $d$ be a prescribed embedding dimension. Our goal in this paper is to learn a judicious parametric mapping $\mathbf{\Phi}$ that predicts the nodal embeddings $\hat{\mathbf{X}} = \mathbf{\Phi}(\mathbf{A}; \mathcal{H}) \in \mathbb{R}^{N \times d}$ in (1) by minimizing the empirical risk

$$\mathcal{L}(\mathcal{H}) := \frac{1}{T} \sum_{i \in \mathcal{T}} \ell(\mathbf{A}^{(i)}, \mathbf{\Phi}(\mathbf{A}^{(i)}; \mathcal{H}) \mathbf{\Phi}^{\top}(\mathbf{A}^{(i)}; \mathcal{H})) \tag{4}$$

to search for the best parameters $\mathcal{H}$. The loss $\ell$ can be adapted to the task at hand. In most cases we will adopt the squared reconstruction error discussed so far, where $\ell(\mathbf{A}, \mathbf{\Phi}(\mathbf{A}; \mathcal{H}) \mathbf{\Phi}^{\top}(\mathbf{A}; \mathcal{H})) := \left\| \mathbf{M} \circ (\mathbf{A} - \mathbf{\Phi}(\mathbf{A}; \mathcal{H}) \mathbf{\Phi}^{\top}(\mathbf{A}; \mathcal{H})) \right\|_{F}^{2}$ [cf. (1)]. In cases where we would like to learn *discriminative* embeddings that are suitable for a downstream task (e.g., node or link prediction), it is prudent to augment the reconstruction error with a task-dependent loss function such as cross-entropy. Such integration of $\Phi$ as a trainable module within a larger (semi-)supervised GRL pipeline will be considered in Section 3.4, where the training set $\mathcal{T}$ will also incorporate task-dependent supervision signals (e.g., node labels).

Next, we discuss the design of the parametric model $\mathbf{\Phi}$ using algorithm unrolling. Architectural refinements are outlined in Section 3.3.

## 3.2 Learned ASE via algorithm unrolling: Gradient descent as neural architecture blueprint

Given the impossibility of a GNN to produce useful embeddings in some important settings discussed in Section 2.2, let us re-consider the original problem in (1). Instead of estimating the ASE through a spectral decomposition of the adjacency matrix [which in fact solves (1) when $\mathbf{M} = \mathbf{1}_N \mathbf{1}_N^{\top}$ as argued in Section 2.1], we could try to solve the optimization problem using iterative methods. This approach was explored in (Fiori et al., 2024), where factored GD was advocated and successfully used to estimate node embeddings, even in online settings where graph data arrive in a streaming fashion and edge status may be partially unknown (as encoded in $\mathbf{M}$).

Let us now briefly present this idea and illustrate how the resulting algorithm lends itself naturally to a re-interpretation as a GNN through the framework of algorithm unrolling we discussed in Section 2.3. For ease of exposition, we will not consider the mask $\mathbf{M}$ here, and leave its inclusion to the next section along with other architectural refinements. Let $f : \mathbb{R}^{N \times d} \mapsto \mathbb{R}$ be the smooth, nonconvex objective function $f(\mathbf{X}) = \|\mathbf{A} - \mathbf{X}\mathbf{X}^{\top}\|_{F}^{2}$ to be minimized. The factored GD algorithm consists of the following iterations

$$\mathbf{X}_{l+1} = \mathbf{X}_l - \alpha \nabla f(\mathbf{X}_l), \quad l = 1, 2, \dots, \tag{5}$$

where $\nabla f(\mathbf{X}) = 4(\mathbf{X}\mathbf{X}^{\top} - \mathbf{A})\mathbf{X}$ and $\alpha > 0$ is the step size. Given the favorable landscape of $f(\mathbf{X})$, the local and global convergence properties of (5) are well understood (Bhojanapalli et al., 2016; Chi et al., 2019; Vu & Raich, 2021; Fiori et al., 2024). In practice, a randomly initialized $\mathbf{X}_0$ will suffice for convergence to an element $\hat{\mathbf{X}}$ in the set of global minimizers [recall from Section 2.1 that the solution of (1) is not unique due to rotational ambiguities]. Substituting the expression of the gradient $\nabla f(\mathbf{X})$ in (5), we arrive to the following iterative update rule

$$\begin{aligned} \mathbf{X}_{l+1} &= \mathbf{X}_l - 4\alpha(\mathbf{X}_l \mathbf{X}_l^{\top} - \mathbf{A})\mathbf{X}_l \\ &= (\mathbf{I}_N + 4\alpha\mathbf{A})\mathbf{X}_l - 4\alpha \mathbf{X}_l \mathbf{X}_l^{\top} \mathbf{X}_l. \end{aligned} \tag{6}$$

Note that the first summand in (6) is a first-order graph convolution as in (2), where the graph shift operator is the adjacency matrix $\mathbf{A}$, and the filter coefficients are $\mathbf{H}_0 = \mathbf{I}_d$ and $\mathbf{H}_1 = 4\alpha\mathbf{I}_d$. In turn, the second summand

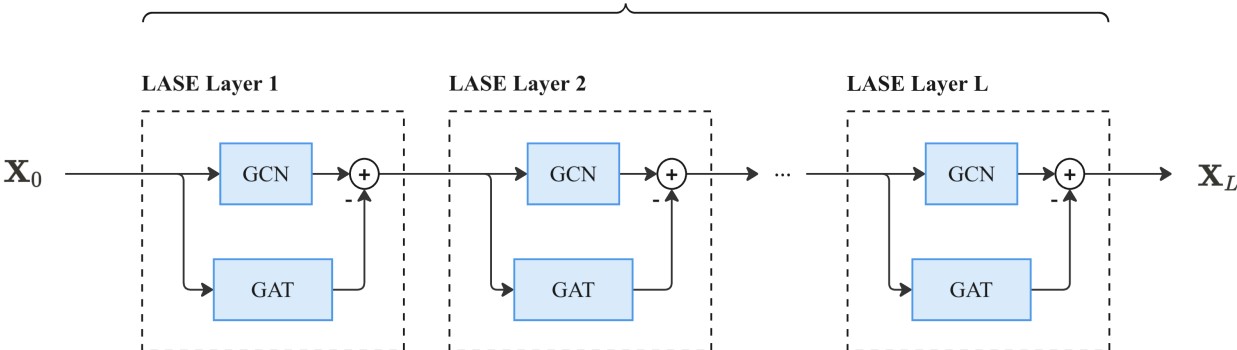

Figure 3: An illustration of the proposed LASE unrolled NN, where $L$ LASE blocks are seamlessly stacked, equivalent to truncating the GD algorithm (6) to $L$ iterations. Each LASE block consists of the superposition of a GCN with a GAT module, the latter acting on a fully-connected graph and using an inner-product-based attention mechanism. Input $\mathbf{X}_0$ consists of random samples, the same initialization as in the GD algorithm, resulting in estimated spectral embeddings $\hat{\mathbf{X}} = \Phi(\mathbf{A}; \mathcal{H}) := \mathbf{X}_L$, where $\mathcal{H}$ includes the learnable weights of all the GCN and GAT modules.

can be interpreted as a Graph Attention Network (GAT) (Veličković et al., 2018) in a fully-connected graph, with a particular choice of attention coefficients and no pointwise non-linear activation function. Recall that in a GAT, the output of the next layer is similar to a GNN, except that the entry $i, j$ in the graph shift operator (i.e., the so-called attention coefficient) now depends on the signal on both incident nodes $\mathbf{x}_{l,i}$ and $\mathbf{x}_{l,j}$. In this case, attention coefficients are collected in the matrix $4\alpha \mathbf{X}_l \mathbf{X}_l^\top$. That is to say, they are expressed as an inner-product without the typical softmax used in (Veličković et al., 2018). While admittedly the attention mechanism is not employed in the standard GAT manner, in our view the GAT reference serves as an analogy to help better contextualize our method within the broader landscape of GNNs.

Taking this into consideration and letting some of the coefficients to be learnable (matrices $\mathbf{H}_1$ and $\mathbf{H}_2$ below), each GD iteration in (6) can be reinterpreted as an NN layer. The $l$-th layer, which we will generically denote as the Learned Adjacency Spectral Embedding (LASE) block, produces the following output:

$$\mathbf{X}_{l+1} = \mathbf{X}_l + \mathbf{A}\mathbf{X}_l\mathbf{H}_1 - \mathbf{X}_l\mathbf{X}_l^\top\mathbf{X}_l\mathbf{H}_2 \quad \Rightarrow \quad \mathbf{X}_{l+1} = \mathbf{GCN}(\mathbf{X}_l) - \mathbf{GAT}(\mathbf{X}_l), \tag{7}$$

where the GCN term, as the GAT, also omits the non-linearity. Using random samples as $\mathbf{X}_0$ and cascading $L$ of these LASE blocks [thus producing the parametric mapping $\hat{\mathbf{X}} = \Phi(\mathbf{A}; \mathcal{H}) := \mathbf{X}_L$; cf. (4)] can be interpreted as truncating the GD algorithm to $L$ steps. We will refer to the resulting $L$-layer unrolled NN as LASE. This process is illustrated in the diagram in Fig. 3.

The LASE network undergoes self-training across realizations $\mathcal{T} := \{\mathbf{A}^{(i)}\}_{i=1}^T$ of a specific graph distribution, optimizing parameters $\mathcal{H} := \{\mathbf{H}_1, \mathbf{H}_2\}$ to minimize the objective function $\mathcal{L}(\mathcal{H})$ in (4) through gradient-based learning techniques. In Section 4.1 we empirically show that the necessary depth of the trained LASE is orders of magnitude smaller than the number of iterations required by the GD algorithm to achieve convergence. This is significant, because the inference time over multiple (potentially large) tests graphs can be markedly reduced.

**Properties of the LASE architecture.** In the discussion so far we have assumed that the learnable weights $\mathcal{H} := \{\mathbf{H}_1, \mathbf{H}_2\}$ are shared across LASE layers, as consequence of the strict unrolling procedure. This approach has the benefit that we could train a LASE network with $L_{\text{train}}$ layers, but then use it for inference with $L_{\text{test}} > L_{\text{train}}$ layers, potentially improving the estimated spectral embeddings during inference while keeping the training cost at a (resource-dependent) minimum. However, if we decouple the parameters of each layer and learn them independently, we may increase the expressive power of the NN and thus boost its performance for a given depth $L$. In our experiments we have opted for this latter alternative, using a total of $2 \times L \times d^2$ independent parameters in $\mathcal{H} = \{\mathbf{H}_{1,l}, \mathbf{H}_{2,l}\}_{l=1}^L$.

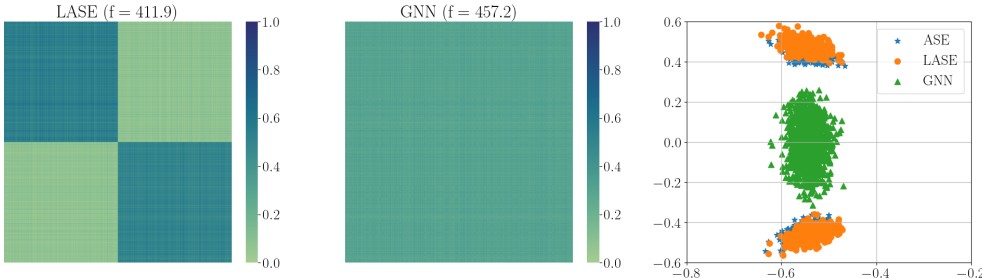

Figure 4: GNN and LASE models are trained with several samples of a symmetric SBM using a loss of the form (1). The left and center panels show the estimated probability matrix $\hat{\mathbf{P}} = \hat{\mathbf{X}}\hat{\mathbf{X}}^\top$ for LASE and the GNN, respectively (the reconstruction error $f = \|\mathbf{M} \circ (\mathbf{A} - \hat{\mathbf{X}}\hat{\mathbf{X}}^\top)\|_F^2$ is also shown for reference). The rightmost panel plots the estimated spectral embeddings $\hat{\mathbf{X}}$ (including ASE as a reference). Whereas LASE using only $L = 5$ layers produces useful spectral embeddings closely matching the ASE, a GNN fails.

Regarding computation cost, note that the number of learnable parameters is, as in all convolutional GNNs, independent of the number of nodes $N$ – and typically much smaller than $N$, as it is customary with unrolled NNs (Monga et al., 2021). However, the greatest bottleneck resides in the GAT module, where the term $\mathbf{X}_l\mathbf{X}_l^\top$ translates to $N^2$ pairwise interactions over a fully-connected graph. This negates one of the benefits of GNNs, which is leveraging the graph's sparse connectivity structure to compute graph convolutions through local, distributed message passing operations. To alleviate this problem, we will implement so-called sparse attention mechanisms; see the ensuing section about architectural refinements for full details.

Besides the alluded computational benefits, that the number of parameters is independent of $N$ also implies that we may train the model with graphs of certain size, and then infer the embeddings on any other graph, including larger ones. Naturally, in order to produce useful embeddings these graphs should stem from a similar underlying distribution, and we will empirically study the generalization capability of LASE in the next section; see e.g., (Ruiz et al., 2020) for theoretical results regarding transferability of GNNs. As a practical benefit of this feature, consider computing node embeddings for a large graph. We contend one can train the model on several relatively small subgraphs of the original one. During inference (less computationally demanding), we compute the embeddings for the large graph; see the results in Section 4.

Finally, recall that GD was initialized at random and for LASE we follow suit. This stands in stark contrast with Example 1, where we showed the inability of GNNs to produce useful spectral embeddings from noisy graph signals (or nodal attributes) in a symmetric SBM. As we argue next, LASE avoids this pitfall and is able to recover embeddings with similar accuracy to ASE or GD, but using fewer iterations (i.e., layers).

**Example 2 (Symmetric SBM revisited)** Going back to Example 1, recall there we considered an SBM graph with $C = 2$ communities of equal size and symmetric inter-class connection probability matrix $\mathbf{\Pi} = \left(\begin{smallmatrix} 0.5 & 0.1 \\ 0.1 & 0.5 \end{smallmatrix}\right)$. To avoid over-fitting the input signal, we may expose the learning architecture to several samples of both the input noise and even the graph (by generating several SBMs with the given $\mathbf{\Pi}$).

For instance, let us consider a 2-layer GNN where each layer consists of a convolution of order $K = 3$ followed by a tanh non-linearity (except in the last layer). All dimensions, including the hidden layers, are $d = 2$. We trained this GNN and LASE (with $L = 5$ layers and $d = 2$) using $T = 100$ graph and noise samples, and a representative inference result (for a new noise and graph sample) is depicted in Fig. 4. As expected, the GNN is unable to produce useful embeddings for these regular graphs, whereas LASE is successful.

## 3.3 Architectural refinements

The bare-bones LASE model presented in the last section can be further customized and enhanced, as traditionally done with unrolled deep learning architectures; see e.g., (Monga et al., 2021). Some architectural refinements are motivated by computational considerations, like the cost of a dense GAT module, and some

others allow the model to operate under more general settings. In the sequel we describe a series of refinements integrated into LASE, with the objective of enhancing the model's robustness and performance.

**Node-wise normalization.** To improve stability during training, node-wise normalization was added to the LASE model. This normalization step consists of dividing both the GCN and GAT terms by the total number of nodes $N$. This action helps maintain numerical quantities at a balanced scale, mitigating any instability that may arise due to, e.g., exploding gradients. The number of nodes is defined as a parameter of the network, making it possible to seamlessly change it during inference. Accordingly, the LASE block architecture introduced in (7) is updated to

$$\mathbf{X}_{l+1} = \mathbf{X}_l + \frac{1}{N}\mathbf{A}\mathbf{X}_l\mathbf{H}_1 - \frac{1}{N}\mathbf{X}_l\mathbf{X}_l^\top\mathbf{X}_l\mathbf{H}_2. \tag{8}$$

**Incorporation of a mask.** For ease of exposition, the mask matrix was not considered during the derivation of the LASE block. To address the original formulation (1), for instance to specify observed and unobserved (or missing) data, the expression for the block computation including the mask $\mathbf{M}_{\text{obs}}$ becomes

$$\mathbf{X}_{l+1} = \mathbf{X}_l + \frac{1}{Np_{\text{obs}}}(\mathbf{M}_{\text{obs}} \circ \mathbf{A})\mathbf{X}_l\mathbf{H}_1 - \frac{1}{Np_{\text{obs}}}(\mathbf{M}_{\text{obs}} \circ (\mathbf{X}_l\mathbf{X}_l^\top))\mathbf{X}_l\mathbf{H}_2, \tag{9}$$

where $p_{\text{obs}}$ is the proportion of present edges with respect to the number of nodes in the mask matrix $\mathbf{M}_{\text{obs}}$.

**Sparse attention mechanisms.** By default, the attention mechanism employed in the GAT term assumes a fully-connected graph, emulating a full attention mechanism with quadratic all-to-all interactions $\mathbf{X}_l\mathbf{X}_l^\top$. To attain higher computational efficiency, we have explored and implemented several sparse attention mechanisms to bring down the quadratic interactions to linear ones. These mechanisms entail various types of (sparse) graphs in the GAT block, including: (i) those generated with an Erdös–Rényi model at different sparsity levels; (ii) those generated with a Watts–Strogatz model; and (iii) the integration of the Big-Bird attention mechanism (Zaheer et al., 2020) commonly used in language transformers. Given the aforementioned considerations, the LASE block is updated as follows

$$\mathbf{X}_{l+1} = \mathbf{X}_l + \frac{1}{Np_{\text{obs}}}(\mathbf{M}_{\text{obs}} \circ \mathbf{A})\mathbf{X}_l\mathbf{H}_1 - \frac{1}{Np_{\text{obs}}p_{\text{att}}}\mathbf{M}_{\text{obs}} \circ (\mathbf{M}_{\text{att}} \circ (\mathbf{X}_l\mathbf{X}_l^\top))\mathbf{X}_l\mathbf{H}_2, \tag{10}$$

where $\mathbf{M}_{\text{att}}$ is the adjacency matrix of the sparse graph used in the GAT term to improve computational efficiency (in other words, a sparse attention mask), and $p_{att}$ is the proportion of edges with respect to the number of nodes in matrix $\mathbf{M}_{\text{att}}$. Note that $\mathbf{M}_{\text{att}}$ is completely unrelated to $\mathbf{A}$, just as in the fully-connected case. The idea is to rapidly propagate the information across all nodes, so for a given sparsity level an $\mathbf{M}_{\text{att}}$ corresponding to a graph with a smaller diameter should yield better results. The performance of the different sparse attention mechanisms (i)-(iii) is reported in Section 4 and a discussion on the inference time-complexity of LASE can be found in Appendix A.5.

**Generalized RDPG.** The vanilla RDPG can only model random graphs with adjacency matrices that are PSD in expectation (recall from Section 2.1 that $\mathbf{P} = \mathbb{E}[\mathbf{A}] = \mathbf{X}\mathbf{X}^\top$). This limitation is overcome by the Generalized RDPG (GRDPG) model (Rubin-Delanchy et al., 2022), which extends the RDPG framework to accommodate a broader range of graph structures, including those with mixed assortative and disassortative behaviors. The idea is to introduce a diagonal matrix $\mathbf{Q}$, with $p$ entries equal to $+1$ and $q$ entires equal to $-1$ (such that $d = p + q$), so that the expected adjacency matrix is now $\mathbb{E}[\mathbf{A}] = \mathbf{X}\mathbf{Q}\mathbf{X}^\top$. The corresponding objective function and estimator are then expressed as [cf. (1)]

$$\hat{\mathbf{X}} \in \underset{\mathbf{X}\in\mathbb{R}^{N\times d}}{\arg\min} \|\mathbf{A} - \mathbf{X}\mathbf{Q}\mathbf{X}^\top\|_F^2. \tag{11}$$

We can mimic the algorithm construction and unrolling processes in Section 3.2. The gradient of the cost function is $\nabla f(\mathbf{X}) = 4(\mathbf{X}\mathbf{Q}\mathbf{X}^\top - \mathbf{A})\mathbf{X}\mathbf{Q}$, and therefore the unrolling of the GD algorithm with step size $\alpha$ is

$$\mathbf{X}_{l+1} = \mathbf{X}_l - 4\alpha(\mathbf{X}_l\mathbf{Q}\mathbf{X}_l^\top - \mathbf{A})\mathbf{X}_l\mathbf{Q} \tag{12}$$

$$= \mathbf{X}_l + 4\alpha\mathbf{A}\mathbf{X}_l\mathbf{Q} - 4\alpha(\mathbf{X}_l\mathbf{Q}\mathbf{X}_l^\top)\mathbf{X}_l\mathbf{Q}. \tag{13}$$

Note that the required modifications to the LASE block are minimal, still consisting of a combination of GCN and GAT modules with minor adaptations, namely

$$\mathbf{X}_{l+1} = \mathbf{X}_l + \mathbf{A}\mathbf{X}_l\mathbf{H}_1\mathbf{Q} - \mathbf{X}_l\mathbf{Q}\mathbf{X}_l^\top\mathbf{X}_l\mathbf{H}_2\mathbf{Q}. \tag{14}$$

Taking $\mathbf{H}_1 = \mathbf{H}_2 = 4\alpha\mathbf{I}_d$ we recover (13). The only difference with respect to (7) is the diagonal matrix $\mathbf{Q}$, and we can recover the legacy LASE block when $\mathbf{Q} = \mathbf{I}_d$. However, if we have reasons to believe that the graph is not fully homophilic, matrix $\mathbf{Q}$ may be considered a hyper-parameter; i.e., for a given $d$, one can verify how many negative values in $\mathbf{Q}$ work best in terms minimizing the the overall reconstruction cost (note that there are only $d$ possibilities). If a spectral decomposition of the adjacency matrix is computationally viable, then a more direct method to estimate the entries of $\mathbf{Q}$ is to use the sign of the $d$ dominant eigenvalues of $\mathbf{A}$. Since computing the eigenvectors of a (potentially large) adjacency matrix could negate the computational benefits of LASE altogether, we can instead compute the $d$ dominant eigenvalues of $m$ smaller random subgraphs sampled from the observed large $G$. This way we obtain a set $\{\mathbf{Q}_1, \ldots, \mathbf{Q}_m\}$ and we choose $\mathbf{Q}$ through the sign of the resulting average $1/m \sum_1^m \mathbf{Q}_m$. We use this latter approach for the experiments in Section 4, but note that several other possibilities exist, such as for instance directly learning the diagonal entries in $\mathbf{Q}$.

### 3.4 End-to-end graph representation learning

So far we have focused on unsupervised learning to approximate spectral graph embeddings with LASE. However, these embeddings are typically used for a downstream task (e.g., node classification) as spectral PEs (Wang et al., 2022). Here we briefly articulate how to use LASE in this context. The first and perhaps most natural option would be to use pre-trained LASE outputs, just like any other PE. That is, first train LASE and then concatenate the predicted PEs $\mathbf{X}_L$ to the input signal $\mathbf{X}_{\text{in}}$ (if nodal attributes are available). A GNN (or whatever architecture we choose) would then produce the mapping between a concatenation $\mathbf{X}_{\text{in}}||\mathbf{X}_L$ to the output (e.g., one-hot encoding of the node's corresponding class). This requires a second training phase, now of the GNN parameters to e.g., minimize a cross-entropy loss.

A second option is to leverage the fact that the LASE block is fully differentiable, making it a viable candidate for inclusion within a larger GRL pipeline designed for some downstream task. Going back to our node classification example using a GNN, instead of *first* adapting the LASE weights $\mathcal{H}$ to minimize equation 1 and *then* training the GNN to minimize a cross-entropy loss, we could instead consider a combined loss and train them jointly. The resulting end-to-end learning system (which we will denote as LASE E2E) is schematically depicted in Fig. 5. It consists of a LASE block, whose output is concatenated with the nodes' features $\mathbf{X}_{\text{in}}$ and fed to a GNN, which will in turn produce the corresponding predictive output $\hat{\mathbf{Y}}$. The system is trained in an end-to-end fashion by minimizing a loss that combines the reconstruction error (1) (for which only the output $\mathbf{X}_L = \mathbf{\Phi}(\mathbf{A}; \mathcal{H}_{\text{LASE}})$ of the LASE module is considered) and the task at hand (for which only the output $\hat{\mathbf{Y}} = \text{GNN}(\mathbf{X}_{\text{in}}||\mathbf{X}_L; \mathcal{H}_{\text{GNN}})$ of the GNN is used). For instance, and again considering the semi-supervised node classification example in a transductive setting where we observe a single graph with adjacency matrix $\mathbf{A}$, the combined loss could take the form

$$\mathcal{L}(\mathcal{H}_{\text{GNN}}, \mathcal{H}_{\text{LASE}}) = \mu\text{CrossEntropy}(\mathbf{Y}, \hat{\mathbf{Y}}) + (1 - \mu)\|\mathbf{M} \circ (\mathbf{A} - \mathbf{X}_L\mathbf{Q}\mathbf{X}_L^\top)\|_F^2, \tag{15}$$

where $\hat{\mathbf{Y}} = \text{GNN}(\mathbf{X}_{\text{in}}||\mathbf{X}_L; \mathcal{H}_{\text{GNN}})$ and $\mathbf{Y}$ are the predicted and actual class labels of each node in the training set, respectively; and $\mathbf{X}_L$ is the output of an $L$-layered LASE block. Furthermore, hyper-parameter $\mu \in [0, 1]$ controls the relative weight between recovering pure spectral PEs ($\mu = 0$) and focusing solely on the task at hand, neglecting the PEs ($\mu = 1$). Our results in Section 4.2 evidence that LASE E2E offers truly learnable PEs, that outperform relevant baselines in node classification and link prediction tasks.

## 4 Experimental Results

In this section we experimentally validate the proposed LASE architecture in two GRL tasks. Firstly, as an alternative of ASE or GD to compute the spectral embeddings of a large graph. In this case, LASE may be trained on smaller sampled sub-graphs including only a fraction of the nodes of the original one. The training set comprises pairs $i \in \mathcal{T}$, each consisting of a graph $\mathbf{A}^{(i)}$ and a random noise sample $\mathbf{X}_0^{(i)}$

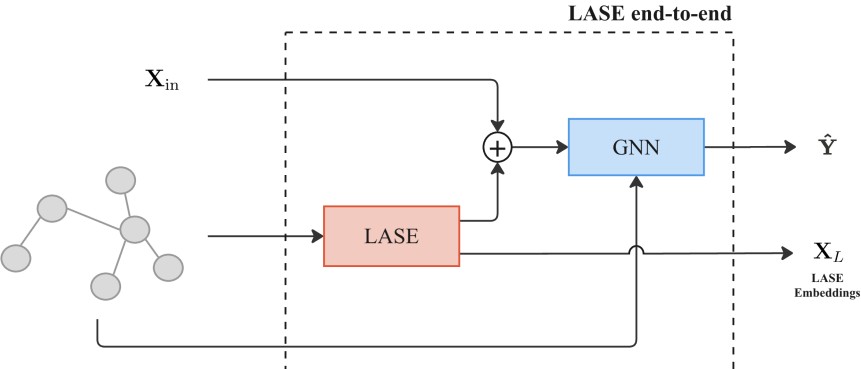

Figure 5: LASE end-to-end (E2E) architecture. The output of an $L$-layered LASE block is concatenated with the node features, which are then processed by a GNN (or whatever variation of this architecture is chosen). The system is trained by minimizing a combined loss that considers both the task at hand through $\hat{\mathbf{Y}}$ as well as the reconstruction through $\mathbf{X}_L$. See (15) for an example of such combined loss.

used as input to LASE. Just as in Example 2, even if we have a single graph we generate several pairs, each including an independently sampled noise. However, when training on smaller subgraphs, each training point contains a randomly sampled subgraph. In any case, the training procedure follows the standard approach of employing mini-batch stochastic gradient descent (SGD) with backpropagation to minimize the empirical risk (4). Regarding the attention mask in (10), unless otherwise stated we use a fully-connected $\mathbf{M}_{\text{att}} = \mathbf{1}_N \mathbf{1}_N^\top$. As we show in Section 4.1, even when using the sub-graph sampling strategy the estimated nodal embeddings on the original graph are of comparable quality to ASE or GD, in terms of the objective function (1). Notably, we demonstrate that LASE offers the added advantages of being: (i) computationally feasible in scenarios where an eigen-decomposition as in ASE is not a viable approach; (ii) markedly faster than GD or its iterative variants; and (iii) able to embed adjacency matrices with missing edge information.

The other task is the one we discussed in Section 3.4, namely using LASE in an end-to-end GRL pipeline, instead of concatenating the spectral embeddings computed by LASE with the nodes' features to be jointly processed by another learning architecture. The latter baseline is a typical procedure adopted when spectral embeddings are used as PEs. Training is similar to the case where we only compute the spectral embeddings, except that we now augment the training set with the corresponding labels $\mathbf{Y}^{(i)}$, and the loss is modified accordingly [recall (15)]. Results in Section 4.2 corroborate that LASE E2E, which endows the embeddings with a discriminative bias towards the task at hand, outperforms relevant baselines.

**Implementation details.** Before moving on, some general remarks regarding implementation details are in order[2]. To compute the ASE we rely on the state-of-the-art RDPG inference library `Graspologic` (Chung et al., 2019). Our implementation of LASE is based on `PyG` (Fey & Lenssen, 2019), fully integrated as a new message passing layer in this popular framework. All experiments were run on a server equipped with an NVIDIA GeForce RTX 3060 (12 GB) GPU and a 13th generation i5-13400F processor with 64 GB of RAM. As input to LASE we use $\mathbf{X}_0$ with i.i.d. random entries sampled from the uniform distribution in [0,1]. This corresponds to the same initialization of the factored GD in (Fiori et al., 2024), for which here we chose the step size $\alpha$ in (5) through the Armijo rule. The embeddings' dimension $d$ is considered a hyper-parameter. For the selection of $\mathbf{Q}$ in GLASE, we rely on the subgraph sampling plus eigendecompoistion method described in the closing discussion of Section 3.3.

---

[2]All anonymized source code and examples are available as supplementary files that accompany this submission.

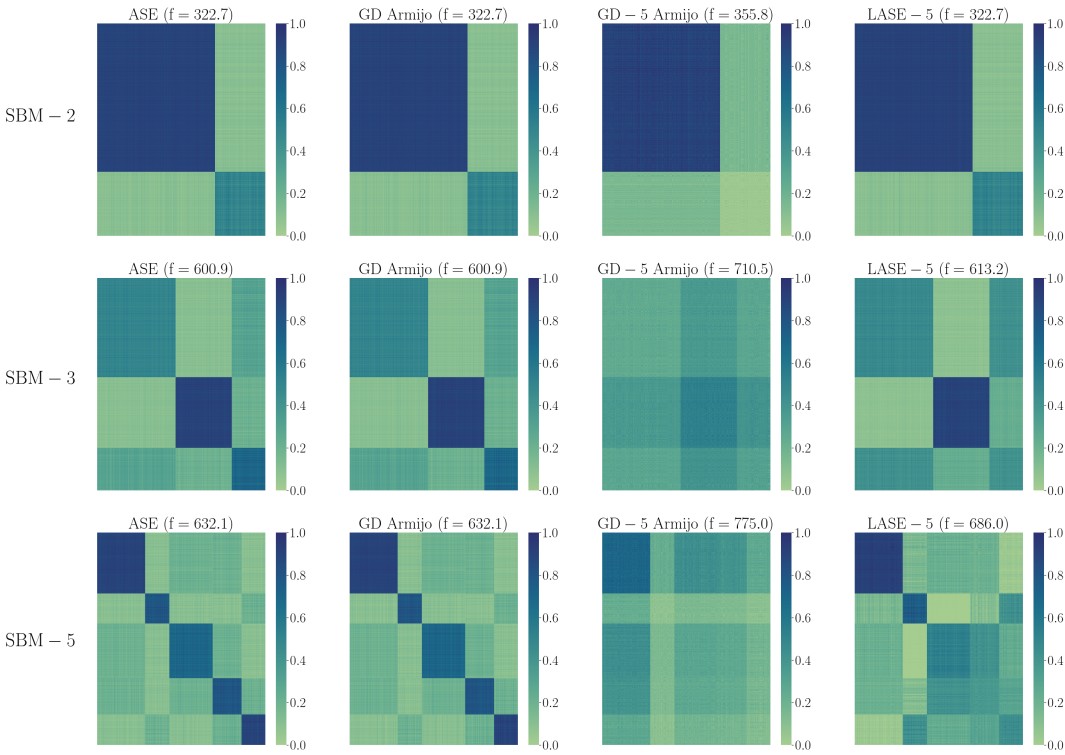

Figure 6: The estimated $\hat{\mathbf{P}} = \hat{\mathbf{X}}\hat{\mathbf{X}}^\top$ using different methods (ASE, GD until convergence, 5 steps of GD and LASE using 5 layers) tested in different synthetic SBM graphs. The resulting reconstruction cost is shown above each estimation. Note how LASE obtains significantly better results than GD when using the same number of iterations.

## 4.1 LASE to compute the spectral embeddings

### 4.1.1 Comparison to baselines

We first turn our attention to the quality of LASE's estimates $\mathbf{X}_L$. In particular, we will compare it to both the eigendcomposition-based gold standard ASE as well as the iterative GD method that served as a blueprint for LASE. Regarding the latter, we implement two variants: running GD until convergence or using the same number of iterations as layers in LASE ($L = 5$).

Let us begin by considering SBM random graphs. As in Example 2, LASE was trained using $T = 1000$ samples of both the input noise and graph adjacency matrices generated with the prescribed $\mathbf{\Pi}$. Inference results in the form of the estimated probability matrix $\hat{\mathbf{P}} = \hat{\mathbf{X}}\hat{\mathbf{X}}^\top$ for the four methods considered (ASE, GD Armijo, GD-5 Armijo, LASE) and corresponding to three different SBMs are depicted in Fig. 6. The main difference between the SBMs considered is the number of nodes ($N = 100, 150$, and $300$) and communities ($C = 2, 3$ and $5$ respectively). For all methods in this test case, we used $d = C$.

As expected, the two leftmost columns of Fig. 6 show that ASE and GD (when run until convergence) yield similar results in terms of the overall reconstruction error $f = \|\mathbf{M} \circ (\mathbf{A} - \hat{\mathbf{X}}\hat{\mathbf{X}}^\top)\|_F^2$ (indicated above each subplot). It is interesting to note that GD Armijo required 44, 38 and 22 iterations, respectively, for each of the graphs tested. If we instead run GD for 5 iterations only, performance markedly degrades, particularly

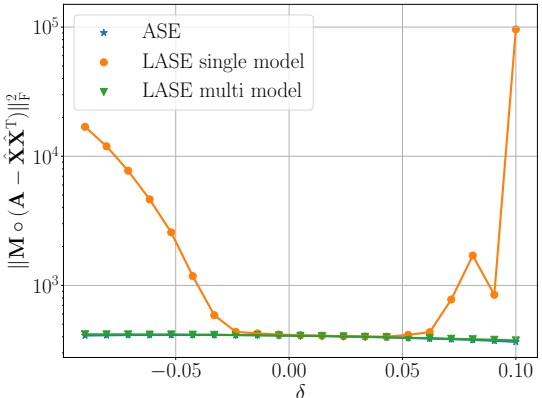 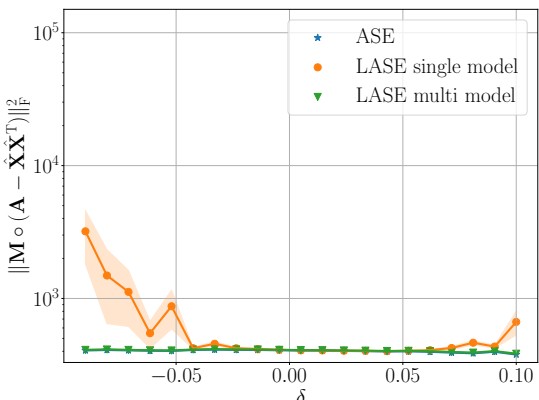

Figure 7: If trained with graphs stemming from an SBM with a given $\mathbf{\Pi}_1$, inference for graphs following a perturbed SBM with $\mathbf{\Pi}_2 = \mathbf{\Pi}_1 + \mathbf{\Delta}$, where $\mathbf{\Delta} = \delta \mathbf{1}_2 \mathbf{1}_2^\top$ (left) and $\mathbf{\Delta} = \left( \begin{smallmatrix} a & b \\ b & c \end{smallmatrix} \right)$ (right, with $a$, $b$ and $c$ following a uniform distribution in $[0, \delta]$) can result in poor performance as $|\delta|$ increases. On the other hand, including all of these models during training results in a remarkably robust LASE, with a performance as good as ASE for all the considered values of $\delta$.

as the number $C$ of SBM communities increases (check the third column in Fig. 6). This is not the case for LASE, which offers embeddings of similar quality as ASE, except in the last and most challenging graph SBM-5. Note, however, that LASE reconstruction are always significantly better than GD-5 Armijo.

### 4.1.2 Robustness to distribution shifts

Note that in the previous examples we have used graphs stemming from the same distribution both for training as well as inference. An intriguing question is how robust is LASE to changes in the underlying distribution. To investigate this out-of-distribution generalization issues, we now present an experiment in which LASE is trained using two approaches. In the first one we consider a single SBM model with $C = 2$ communities, characterized by a connection probability matrix $\mathbf{\Pi}_1$ and $N_1 = 100$, with 70% of the nodes belonging to community 1. Inference is then performed on larger graphs with $N_2 = 1000$ nodes, generated by SBM models with $\mathbf{\Pi}_2 = \mathbf{\Pi}_1 + \mathbf{\Delta}$. The proportion of nodes in each community remains consistent with the training setup. We will consider two types of perturbations $\mathbf{\Delta}$. In the first one, we add a constant to all entries of $\mathbf{\Pi}_1$; i.e. $\mathbf{\Delta} = \delta \mathbf{1}_2 \mathbf{1}_2^\top$, where the perturbation $\delta$ varies within the range $[-0.09, 0.1]$. We also consider a non-uniform perturbation where $\mathbf{\Delta} = \left( \begin{smallmatrix} a & b \\ b & c \end{smallmatrix} \right)$. In this case, the perturbation coefficients $a$, $b$, $c$ are sampled from a uniform distribution in the interval $[0, \delta]$, and $\delta$ also varies within the range $[-0.09, 0.10]$.

Reconstruction-error results corresponding to the uniform (left) and non-uniform (right) case as we vary $\delta$ are shown with orange circles in Fig. 7. Note how the reconstruction cost obtained from this single-model LASE remains very similar to the one obtained by ASE (blue stars in the figure) under small perturbations. However, as the absolute value of the perturbation $\delta$ increases, thus markedly deviating from the original connectivity matrix $\mathbf{\Pi}_1$, the resulting error grows and drifts away from that of ASE.

However, the robustness of LASE can be enhanced by exposing it to graphs stemming from multiple of these distributions (parameterized by $\delta$). The second training approach, which we will denote as LASE multi model, includes multiple SBM models using $\mathbf{\Pi}_2$ and $N_1 = 100$, with the same per-class proportion as before. Specifically, the training set includes 500 samples for each $\mathbf{\Pi}_2$, with $\delta$ taking 11 evenly spaced values within the range [-0.09, 0.09], for a total of $T = 5500$ graph samples. The green triangles in Fig. 7 show how, in this case, LASE exhibits a marked reduction in cost, matching ASE's performance for all values of $\delta$.

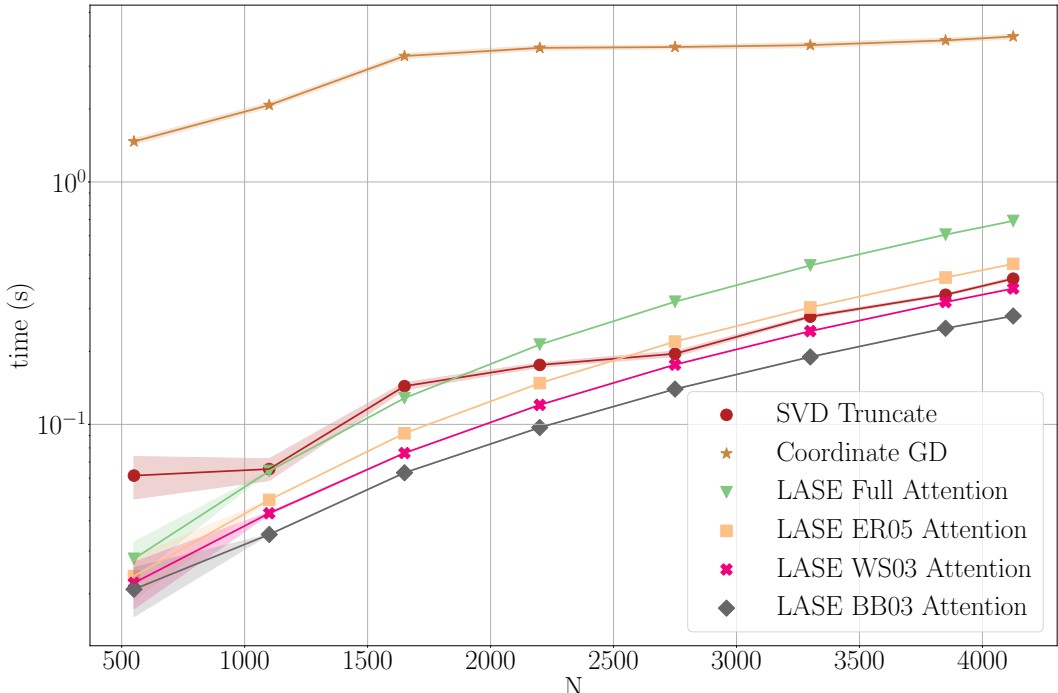

Figure 8: Mean wall-clock time for computing ASE, Block Coordinate Descent (a faster version of GD studied in Fiori et al. (2024)) and LASE using full or sparse attention. This last method runs fastest overall, even outperforming the highly-optimized `Graspologic` routines used for the ASE eigendecomposition.

### 4.1.3 Spectral embeddings in large graphs

Consider a scenario involving a graph so large that computing the eigendecomposition to form the ASE becomes impractical. For instance, a general-purpose PC may struggle to compute the ASE of a moderately-sized graph with approximately $N = 10,000$ (or more) nodes. In such cases, it is essential that LASE's computational cost scales better with $N$. This cost primarily depends on two parameters: (i) the number of layers $L$, which represents the total number of GD iterations we are willing to perform; and (ii) the size of the graph $N$, the central focus of our current discussion.

**Training on smaller sampled subgraphs.** Leveraging the transferability of LASE, we aim at training LASE on smaller portions of the graph that fit within the available computational resources. To illustrate this approach, consider an SBM graph with $N = 12,000$ nodes divided into $C = 3$ communities. Instead of training LASE on the entire graph, we randomly select 5% of the nodes and use the induced subgraph for training. A training set $\mathcal{T}$ including $T = 800$ such samples is generated, and the LASE weights are trained to minimize the average cost across all samples; see (4). In this case, the resulting reconstruction error $\|\mathbf{M} \circ (\mathbf{A} - \hat{\mathbf{X}}\hat{\mathbf{X}}^\top)\|_F^2$ on the original graph is 4296 for ASE, while LASE trained using this subgraph sampling approach achieves a cost of 4498 – a marginal increase of less than 5%. This stands in stark contrast with the results we discussed in Section 2.2, underscoring the importance of transferability in the LASE architecture, which enables the reuse of trained weights for inference on the original large graph.

**Sparse attention.** The GAT sub-module of LASE may still constitute a significant computation bottleneck, since it operates using a fully-connected graph linking all nodes. As discussed in Section 3.3, the literature on transfomers addresses this issue by considering so-called sparse attention; i.e., randomly selecting some edges

| dataset | $N$ | $d$ | ASE Loss | GD Loss (*) | GD-5 Loss | LASE-5 Loss |
|---------|-----|-----|----------|-------------|-----------|-------------|
| Cora | 2708 | 6 | **101.28** | $103.50 \pm 0.01$ | $126.87 \pm 0.01$ | $102.83 \pm 0.01$ |
| Citeseer | 3327 | 6 | 98.17 | **96.07** $\pm 0.01$ | $148.54 \pm 0.01$ | $102.14 \pm 0.02$ |
| Twitch ES | 4648 | 7 | 345.95 | $337.54 \pm 0.02$ | $356.13 \pm 0.01$ | **334.00** $\pm$ **0.03** |
| Amazon | 7650 | 8 | 469.74 | $474.15 \pm 0.01$ | $502.56 \pm 0.02$ | **454.23** $\pm$ **0.02** |

Table 1: The resulting loss $\|\mathbf{M} \circ (\mathbf{A} - \hat{\mathbf{X}}\hat{\mathbf{X}}^\top)\|_F^2$ of ASE, GD, GD using only 5 iterations and a 5-layered LASE when applied to four real-world networks. LASE systematically performs competitively, and as the network size increases it even outperforms the rest of the methods. The standard deviation of the results are also indicated for GD and LASE, illustrating their robustness to the random input signal.

of the fully-connected graph, which may be regarded as another mask [cf. (10)]. There are several choices for $\mathbf{M}_{\text{att}}$, the most popular being Erdös-Rényi (ER; i.e., uniformly sampling edges with a certain probability), Watts-Strogatz (WS; i.e., starting from a regular lattice with a given degree $r$, then randomly rewiring edges with a certain probability $p$) and Big-Bird (BB; i.e., a combination of ER and a regular lattice).

We test these sparsifying mechanisms on an SBM graph with $C = 10$ communities and varying number of nodes $N \in [500, 4000]$. Parameters of the sparse attention approaches were chosen so that the resulting sparsity $p_{\text{att}} \in \{0.1, 0.3, 0.5\}$. For each mechanism we report the time corresponding to the smallest $p_{\text{att}}$ that resulted in a total loss that is within 5% of the one obtained with full attention (itself, and like in the previous sections, almost identical to the one obtained through traditional ASE). This resulted in a mean degree of half of the original fully-connected graph for the ER mechanism, whereas only a third was enough for both WS and BB. Accordingly, we find that pure randomness is detrimental for attention.

Inference times are reported in Fig. 8. Each point in the graph is the mean wall-clock time over 10 independent trials, and the bands around the curves indicate one standard deviation. When using sparse attention mechanisms, inference time is halved with respect to full attention. Furthermore, we have included two references for comparison. Firstly, a variant of the GD method introduced in (Fiori et al., 2024), which uses Block Coordinate Descent and is faster than vanilla GD for this node embedding problem. Note how LASE is roughly an order of magnitude faster than even this variant of GD. Secondly, we have also included the computation time for ASE, estimated through `Graspologic`, which uses highly-optimized routines for the eigen-decomposition of the adjacency graph (SVD Truncate in Fig. 8). Note how either BB or WS (recall that in this case both use a sparsity of 30%) result in faster inference times than this highly-optimized library. We reiterate that in all cases, the total resulting cost is within 5% of the one obtained by `Graspologic`.

### 4.1.4 Embedding real-world networks

Here we asses LASE's ability to approximate node embeddings for real-world graphs. For instance, let us consider a set of networks typically used as benchmarks in several learning tasks: Cora (Yang et al., 2016), Citeseer (Yang et al., 2016), Twitch ES (Rozemberczki et al., 2021), and Amazon Photo (Shchur et al., 2018). The Cora dataset is a collection of $N = 2708$ scientific publications. The connections between these publications correspond to a citation, resulting in 5429 links. The Amazon Photo network is a collection of $N = 7650$ products, where the 238,162 connections signify that the corresponding products are frequently purchased together. The Citeseer dataset is a citation network of $N = 3327$ publications, where nodes represent research papers and the 9104 edges denote citation relationships between them. Finally, the Twitch ES dataset is a network of $N = 4648$ Spanish-speaking Twitch users, where nodes represent gamers and an edge exists if they follow each other (with a total of $|\mathcal{E}| = 123,412$).

We are interested in computing the spectral embeddings of the underlying graphs, and as before we will compare ASE, GD, GD using only 5 iterations, and LASE with $L = 5$. We train LASE using $T = 1000$ sampled subgraphs of the original one, uniformly chosen at random so that approximately 300 nodes are retained in each of them. Full attention is used except for the Amazon network, which used WS attention (with $r = 0.1$ and $p = 0.1$). The resulting reconstruction errors are reported in Table 1. Note how LASE with limited depth performs comparably to the GD algorithm run until convergence. Interestingly, for the two

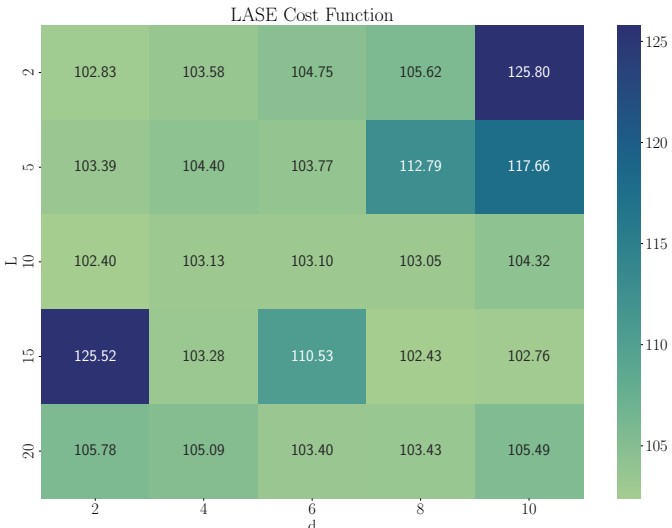

Figure 9: The resulting loss $\|\mathbf{M} \circ (\mathbf{A} - \hat{\mathbf{X}}\mathbf{Q}\hat{\mathbf{X}}^\top)\|_F^2$ of LASE on the Cora dataset, evaluated across different combinations of embedding dimension $d$ and number of layers $L$ (shown in the $x$ and $y$ axis, respectively). Except for few pathological cases, the resulting loss exhibits limited variability with respect to $L$ and $d$.

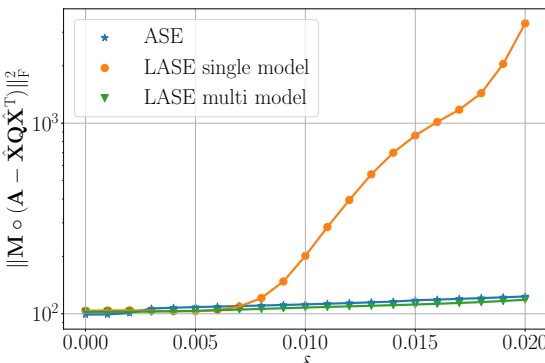

Figure 10: When trained on graphs from a single model (e.g., the Cora dataset), inference on graphs generated by modifying the original structure—through random edge additions and deletions with probability $\delta$—degrades as $\delta$ increases. In contrast, training on a diverse set of such graph models significantly enhances the robustness of LASE, yielding performance comparable to ASE across all tested values of $\delta$.

largest graphs, but specially in the case of Amazon, LASE outperforms GD. The standard deviation of the results are also displayed for GD and LASE, since they both depend on the random initialization/input. The low variability for all of these methods confirms their robustness and ability to attain the global optimum.

**Robustness to hyperparameter selection.** We also examine the effect of the chosen hyperparmeters in the resulting loss. Figure 9 depicts the results of such analysis for the Cora dataset. We have trained and evaluated LASE for a grid of values of the only two hyperparameters of the architecture ($d$ and $L$). Apparently, the model is fairly robust to modifications of these hyperparamter values.

**Robustness to graph perturbations.** Mimicking the experiments we carried out in Section 4.1.2 but now for the real-world datasets, here we verify the robustness of LASE to perturbations on the underlying network. Starting from the Cora dataset, we have randomly added and removed links with a certain probability $\delta$. Results, shown in Fig. 10, are consistent with our previous findings. While inference of LASE as the

| Year | $N$ | $\rho$ | $d$ | ASE | GD | LASE-20 |
|------|-----|--------|-----|------|-----------------|-----------------|
| 1960 | 154 | 0.115 | 4 | 23.98 | $18.00 \pm 0.01$ | $18.81 \pm 0.08$ |
| 1970 | 194 | 0.124 | 4 | 30.89 | $20.79 \pm 0.01$ | $21.85 \pm 0.04$ |
| 1980 | 257 | 0.102 | 4 | 35.17 | $20.71 \pm 0.01$ | $22.04 \pm 0.11$ |
| 1993 | 248 | 0.108 | 4 | 30.10 | $13.56 \pm 0.01$ | $15.35 \pm 0.10$ |
| 2004 | 283 | 0.091 | 4 | 36.34 | $20.98 \pm 0.01$ | $23.06 \pm 0.18$ |
| 2017 | 324 | 0.096 | 4 | 45.96 | $30.08 \pm 0.01$ | $31.64 \pm 0.15$ |

Table 2: The resulting loss, $\|\mathbf{M} \circ (\mathbf{A} - \hat{\mathbf{X}}\mathbf{Q}\hat{\mathbf{X}}^\top)\|_F^2$, for both GD and LASE-20 is evaluated across different years of the UN General Assembly dataset. The proportion of unknown votes in each case is denoted by $\rho$.

perturbation increases yields a noticeable degradation, including graphs stemming from these perturbed distributions in the training set (i.e., multi-model training) results in performance that close matches ASE.

**Missing edge data.** Finally, let us consider the problem of embedding a network with unknown edges. To this end, we use United Nations (UN) General Assembly voting data (Voeten et al., 2009). This dataset contains every roll call in the history of the UN General Assembly, including which of the member countries were present (or not) during its voting session, and which was the country's vote (either 'Yes', 'No', or 'Abstain'). We thus construct a graph for each year, where a node represents a roll call or a country. A link between two nodes is included if the corresponding country voted affirmatively for the corresponding roll call; i.e., we build a bipartite graph connecting countries and roll calls. When a country was absent or abstained from voting, we will consider that the corresponding edge is unknown. The evaluation of $\|\mathbf{M} \circ (\mathbf{A} - \hat{\mathbf{X}}\mathbf{Q}\hat{\mathbf{X}}^\top)\|_F^2$ corresponding to the estimation of $\hat{\mathbf{X}}$ for ASE, GD and a LASE architecture with 20 layers, corresponding to six years across several decades, is shown in Table 2 (see Appendix A.4 for further details of the graph for each year). The column denoted by $\rho$ indicates the proportion of unknown edges (from all potential votes) in the given graph. Note how LASE exhibits similar performance to GD, while ASE cannot recover accurate embeddings. Indeed, ASE's total loss is typically 50% over that of GD.

## 4.2 End-to-end graph representation learning for link prediction and node classification

Having shown the computational advantages of LASE over GD or ASE during inference, let us now assess the performance of LASE E2E. Its merits will be evidenced in two scenarios. Firstly, when PEs are required, LASE E2E attains the same (or better) performance than if the PEs were precomputed and used along the node attributes, thus effectively rendering the precomputation step unnecessary. Secondly, if the graph has a non-negligible amount of unobserved edges (typical in several applications, such as recommender systems or the UN test case we study in Section 4.2.1), the ability of LASE E2E to accommodate those also translates to performance gains that may be significant. In the sequel, we illustrate these advantages in two widespread tasks using real-world datasets: link prediction and node classification.

### 4.2.1 Link prediction

We return to the UN General Assembly voting data, where we now include nodal signals (attributes) $\mathbf{X}_{\text{in}}$. Regarding the roll calls, the publishers of the dataset have classified them into six categories (proposals related to the Palestinian conflict, colonialism, human rights, etc.). The signal on each node will be a 12-dimensional one-hot encoding indicating in the first six dimensions the continent to which the country belongs to, and in the rest the category of the roll call.

We consider six randomly selected countries for which we have randomly chosen 30% of the roll calls (among those it voted) for prediction. They are thus also tagged as unknown in the mask matrix $\mathbf{M}_{\text{obs}}$. For the LASE E2E we use a LASE block (with $L = 10$ and $d = 4$), the output of which is concatenated to the node attributes $\mathbf{X}_{\text{in}}$, which are then processed by a two-layer GNN to produce link predictions $\hat{\mathbf{A}}$ (see Fig. 11). The combined loss utilized for this link prediction task is given by

$$\mathcal{L}(\mathcal{H}_{\text{GNN}}, \mathcal{H}_{\text{LASE}}) = \mu \text{CrossEntropy}(\mathbf{A}, \hat{\mathbf{A}}) + (1 - \mu)\|\mathbf{M} \circ (\mathbf{A} - \hat{\mathbf{X}}\mathbf{Q}\hat{\mathbf{X}}^\top)\|_F^2, \tag{16}$$

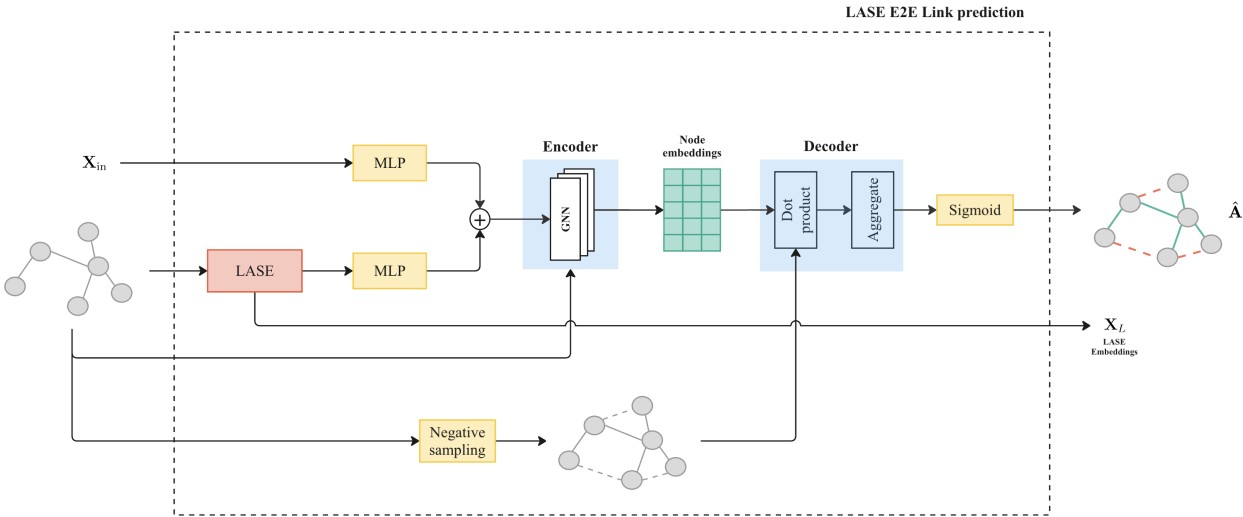

Figure 11: LASE E2E architecture for link prediction. Instead of training LASE separately from the link prediction task, we train a LASE module along with a GNN so that a loss combining both reconstruction and prediction errors is minimized.

To assess the benefits of the E2E architecture, we compare against more traditional way of using spectral PEs; i.e., concatenating precomputed spectral PEs obtained from ASE or LASE to the input signal $\mathbf{X}_{\text{in}}$. We have also included the results when not using PEs at all; i.e., using only the 12-dimensional one-hot encoding signal as input to the GNN. Finally, we also compare against the state-of-the-art approach *PowerEmbed* (Huang et al., 2022); see Section 5 for a detailed discussion of this and other related methods.

Figure 12 shows the results for the same years we considered in Table 2 in the form of a boxplot of the resulting link-prediction accuracy. Each datum in the boxplot corresponds to one of ten experiments (where we have randomly picked the countries and the roll calls to be predicted). The first thing to observe is that the incorporation of spectral PEs by itself leads to a considerable improvement in prediction accuracy in several years (e.g., 1970 or 2004). This is in line with the discussion in Section 2.2, illustrating in a real-life problem the need for PEs and the inability of GNNs to produce this information under certain inputs. Each such experiment includes different countries (and hence votes), so one expects a non-negligible variability in the results. Some countries' voting patterns are aligned with allies (in which case the spectral PEs should be quite informative), whereas others are more isolated and their votes will be harder to predict.

The second aspect to highlight in Fig. 12 is that in several years, using ASE as a PE results in an accuracy which may be significantly less than both LASE PE or its E2E variant; see in particular the years 1993 or 2017. PowerEmbed exhibits similar performance to ASE for these years. This is somewhat expected since, as explained in Section 5, PowerEmbed is based on the power method to compute the dominant eigenvectors, although it also includes an Inception network (Szegedy et al., 2015) which considers all the power iterations in its predictions. In any case, these results illustrate the importance of properly considering unknown edges in the model when computing PEs, an impossibility for ASE that requires the eigendecomposition of a fully-observed adjacency matrix. It is interesting that for the year 1980, where using the PEs or not results in a similar accuracy, the method that performs best is PowerEmbed. This may be indicative that the actual graph structure is not as important as in the rest of the years (while the input signal is informative enough), and the Inception layer learns to ignore the higher-order powers. This observation is further explored in Section 4.2.2, when we shift focus to node-level prediction tasks.

The final noteworthy conclusion that can be drawn from Fig. 12 is that LASE E2E obtains competitive results when compared to LASE as a precomputed PE, even surpassing it in some instances (e.g., the years 1993 or 2004). We remark that for the E2E variant a single training is necessary, resulting in less computational overhead and without sacrificing overall performance.

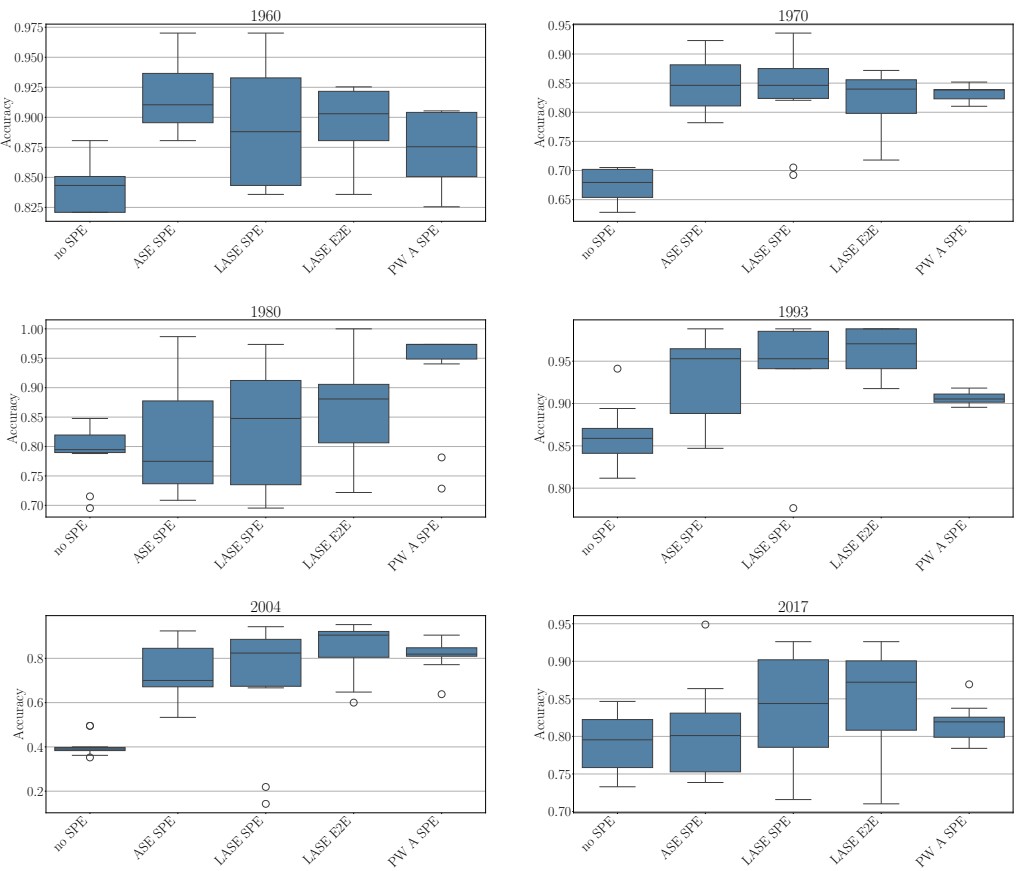

Figure 12: Accuracy in predicting the votes during six years of randomly chosen countries of the UN General Assembly. The evaluated methods include not using PEs at all, using either ASE or LASE as spectral PEs, the E2E architecture discussed in Section 3.4, and the PowerEmbed method (Huang et al., 2022). Not using a PE typically exhibits the worst performance, as verified for all years except 1980. The E2E architecture typically outperforms all other methods for the rest of the years (e.g. 1993, 2004 and 2017), or is at least very competitive. This performance gain comes at a computational cost similar to that of using no spectral PEs, since no pre-computations are required.

### 4.2.2 Node classification

Moving on to node classification, we evaluated the performance of LASE E2E by considering the same four real-world network datasets in Section 4.1.4: Cora, Citeseer, Twitch ES and Amazon. We now also include node signals $\mathbf{X}_{in}$ for each network, and the goal is to predict the category for each node. For Cora, node features consist of a 1433-dimensional one-hot word vector, its entries indicating the presence or absence of a certain word in the publication, and each publication is categorized into one of seven classes. In Amazon, node features also consist of a one-hot word vector encoding product reviews, in this case of 745 dimensions indicating the presence or absence of a certain word. Each product is assigned a label corresponding to one of eight product categories. In the Citeseer network, node features are represented as 3703-dimensional one-hot word vectors, indicating the presence or absence of specific words in a paper's content, with each paper categorized into one of six predefined classes. Finally, the Twitch ES dataset encodes node features as

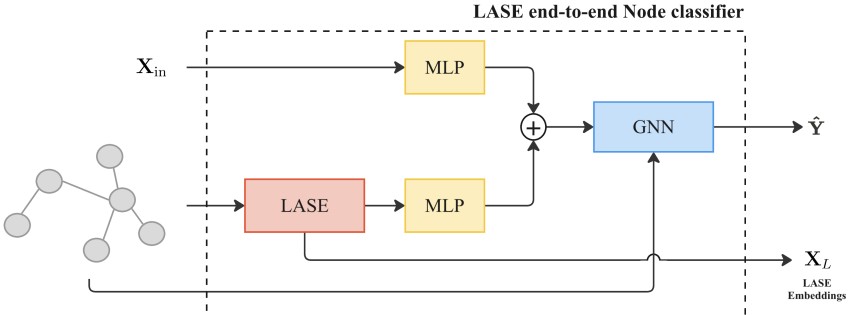

Figure 13: LASE E2E architecture for node classification. The output of a LASE block is concatenated to the node features, but crucially the system is trained end-to-end using a loss that encompasses both reconstruction and classification.

| $\rho$ | No PE | ASE PE | LASE PE | LASE E2E | PowerEmbed RW-10 |
|---|---|---|---|---|---|
| 0 | $86.8 \pm 0.4$ | $86.5 \pm 0.6$ | $86.4 \pm 0.3$ | $\mathbf{86.9 \pm 0.3}$ | $85.0 \pm 0.4$ (*) |
| 0.2 | $84.6 \pm 0.3$ | $85.0 \pm 0.5$ | $85.8 \pm 0.3$ | $\mathbf{85.8 \pm 0.6}$ | $84.2 \pm 0.5$ |
| 0.4 | $83.1 \pm 0.3$ | $83.2 \pm 0.5$ | $84.3 \pm 0.4$ | $\mathbf{85.4 \pm 0.5}$ | $82.7 \pm 0.4$ |
| 0.6 | $79.3 \pm 0.5$ | $79.0 \pm 0.3$ | $83.2 \pm 0.6$ | $\mathbf{85.5 \pm 0.6}$ | $81.6 \pm 0.5$ |
| 0.8 | $73.8 \pm 0.8$ | $73.0 \pm 0.6$ | $79.6 \pm 0.4$ | $\mathbf{84.6 \pm 0.5}$ | $79.1 \pm 0.7$ |

Table 3: Mean accuracy $\pm$ standard deviation over 10 data splits for the CORA network with different proportions of missing edges $\rho$. The results with (*) corresponds to PowerEmbed(Lap)-10 taken from Huang et al. (2022) which uses the Laplacian matrix and 10 iterations.

dense 128-dimensional embeddings that capture information about games liked, user location, and streaming habits, and the primary (binary classification) task is to predict whether a user streams mature content.

Here, the E2E LASE architecture consists of a $L = 5$-layer LASE block (with dimensions $d = 6$ for CORA, $d = 8$ for Amazon, $d = 6$ for Citeseer, and $d = 7$ for Twitch), whose output is concatenated with the node attributes $\mathbf{X}_{\text{in}}$. These were then processed by a three-layer GAT to produce node classifications through a softmax function (see Fig. 13). The resulting model was trained to predict among the different classes by minimizing the combined loss function given in (15), which includes a cross-entropy and reconstruction loss (1), weighted by a hyper-parameter $\mu$. In terms of baselines, we will again consider a GNN with no PEs as well as using precomputed PEs obtained through ASE, a pre-trained LASE and PowerEmbed.

Table 3 reports the results on the Cora dataset. In particular, each row corresponds to a different proportion of missing edges (indicated in the first column as parameter $\rho$). Note how LASE E2E systematically obtains the best results, above PowerEmbed with $\rho = 0$ even with up to 60% of unknown edges. Using pretrained LASE PEs also yields competitive results, since it can account for unknown edges unlike the vanilla ASE. This explains the latter's rapid degradation as $\rho$ increases. We also find that not using any PE performs similarly to using ASE. Again, as the number of unknown edges increases and thus the structural information is progressively lost, the capacity of LASE to provide good approximations of the actual PEs plays a key role in its consistent performance. As a further verification on the importance of the connectivity in the problem at hand, we trained a 2-layered fully connected NN with hidden dimension of 64, using only the node features as input. The resulting performance is $75.2 \pm 0.6$, significantly lower than the rest of the evaluated methods.

Let us now turn our attention to the Amazon dataset. Table 4 shows the same comparison as before. Again, LASE E2E obtains the best results overall, even if it is using sparse attention (in this case a WS with $r = 0.1$ and $p = 0.1$). However, it is interesting to highlight that the degradation in performance as $\rho$ increases is not as significant as before – both for ASE and when we do not use any PE. This suggests that the actual structure of the graph is not as important as the input features. This is verified when we train a simple fully-

| $\rho$ | No PE | ASE PE | LASE PE | LASE E2E | PowerEmbed RW-10 |
|---|---|---|---|---|---|
| 0 | $94.3 \pm 0.6$ | $93.9 \pm 0.6$ | $94.2 \pm 0.6$ (*) | $\mathbf{94.4 \pm 0.4}$ (*) | $94.2 \pm 0.2$ (**) |
| 0.2 | $94.2 \pm 0.4$ | $94.0 \pm 0.5$ | $93.7 \pm 0.8$ (*) | $\mathbf{94.3 \pm 0.6}$ (*) | $94.1 \pm 0.2$ |
| 0.4 | $93.9 \pm 0.7$ | $93.6 \pm 1.5$ | $93.8 \pm 0.6$ (*) | $\mathbf{94.0 \pm 0.3}$ (*) | $93.8 \pm 0.2$ |
| 0.6 | $93.5 \pm 0.6$ | $93.5 \pm 0.5$ | $93.1 \pm 0.9$ (*) | $\mathbf{94.0 \pm 0.6}$ (*) | $93.7 \pm 0.2$ |
| 0.8 | $92.9 \pm 0.7$ | $92.9 \pm 0.6$ | $92.6 \pm 0.7$ (*) | $\mathbf{93.3 \pm 0.6}$ (*) | $93.0 \pm 0.2$ |

Table 4: Mean accuracy $\pm$ standard deviation over 10 data splits for the Amazon Photo network with different proportions of missing edges $\rho$. The results marked with (*) were trained using sparse attention WS with $r = 0.1$ and $p = 0.1$ instead of *full attention*. The results with (**) corresponds to PowerEmbed taken from Huang et al. (2022) which uses the Random Walk matrix and 2 iterations.

connected NN model with 2-layers (with a hidden dimension of 64) for this dataset. This network-agnostic architecture obtains a mean accuracy of $91.7 \pm 0.2$. A similar behavior was verified for both the Citeseer and Twitch ES, whose results are reported in Appendix A.3

## 5 Related Work

**Leveraging the power method to learn spectral embeddings.** One set of works related to ours aims to compute spectral embeddings through GNN iterations that can express the dominant eigenvectors of the graph shift operator. The *PowerEmbed* method (Huang et al., 2022) is based on a spectral decomposition of the adjacency matrix, similar to the RDPG model in this paper, but only taking into account the eigenvectors. Inspired by the power method that is known to converge to the dominant eigenvectors, Huang et al. (2022) propose a normalization step after each GNN iteration, resembling the mentioned power method. The processed outputs of each layer are then concatenated and used for downstream node classification tasks [a so-called Inception layer (Szegedy et al., 2015)]. Since the method is initialized with the node features, initial layers retain this local information, while the final layers capture global spectral properties of the graph as the power iteration converge. A follow-up related paper inspired is (Eliasof et al., 2023), which instead of initializing the model with node features as in *PowerEmbed*, it uses random values. The procedure is similar, alternating a normalization step after some iterations. Eliasof et al. (2023) also propose to learn a graph-dependent propagation operator. Unlike these approaches that are inspired by the power method, the LASE architecture mimics GD iterations through an algorithm unrolling principle.

**On the expressive power of GNNs.** The expressive power of GNNs has been the subject of much research interest over the last few years. In particular, useful links drawn between the classic Weisfeiler-Leman graph isomorphism test and the message-passing GCN have shed light on the incapacity of this class of GNNs to distinguish between some graphs which are structurally different (Xu et al., 2018). These (and other related) findings have motivated a host of alternative GNN architectures with increased expressive power; see e.g., (Sato, 2020; Morris et al., 2024) and references therein.

Instead of modifying the architecture, an alternative to remedy the GCN's inability to distinguish between some non-isomorphic graphs is to endow each node with additional signals (i.e., features or attributes) to break structural symmetries leading to ambiguities. Early works advocated using a unique identifier for each node, via one-hot encoding or a randomly generated label. While simple, these schemes seriously limit the transferability of the learned architecture to other graphs, and may lead to functions that are not equivariant to node permutations (Vignac et al., 2020). This has motivated the use of PEs (Srinivasan & Ribeiro, 2020; Rampášek et al., 2022), some of which we have already discussed in the preceding experiments. An alternative to these spectral-based PEs, which may be interpreted as capturing global information of the graph, is to use methods based on sub-graphs, such as encoding an ego-net around each node (Zhao et al., 2022; Bouritsas et al., 2023). Substructure-based methods offer an attractive trade-off due to their moderate computational and memory requirements, plus they have been shown to effectively address certain limitations of approaches that rely on random signal injection. A detailed comparison with subgraph-based methods

in end-to-end tasks remains an interesting direction for future work, particularly to explore the potential benefits of combining both local and spectral PEs.

On a related note, we comment on the implications of the anonymity of the input signal on a GNN's ability to discriminate nodes. Specifically, as discussed in (Kanatsoulis & Ribeiro, 2024), to evaluate the ability of a GNN to differentiate nodes, one should feed the network with a signal that is not discriminative by itself. Otherwise, one will not be able to tell if the GNN can solely produce such a discriminative output. Kanatsoulis & Ribeiro (2024) prove that under mild assumptions, a GNN with anonymous input produces different outputs for two graphs with different spectral properties. This is somehow related to our analysis in Section 2.2, but in their work, the authors discuss this phenomenon for *two different graphs*, while we focus the analysis for a given graph, distinguishing *nodes* in it. Although these interesting ideas could be adapted to our case, as we show in Appendix A.2 there are pitfalls with, e.g., symmetric SBMs. Another related work, but also devoted to distinguishing graphs, is (Puny et al., 2020). Therein, an attention mechanism is added to the Random GNN (Sato et al., 2021), and it is proved that this aligns with the 2-Folklore Weisfeiler-Lehman algorithm.

**Graph auto-encoders.** Since LASE computes node embeddings from which it is possible to approximately reconstruct a given adjacency matrix, we can frame our work in the graph auto-encoder context; see e.g., (Hamilton, 2020) and the comprehensive review in (Chami et al., 2022). The classical (non-probabilistic) graph auto-encoder (Kipf & Welling, 2016) proposes to compute embeddings $\mathbf{X}$ using a GCN with the features and graph structure as inputs (the encoder), and then reconstruct the adjacency matrix via an outer product decoder $\mathbf{X}\mathbf{X}^\top$, which resembles the RDPG model at the heart of our approach. Follow-up works such as (Ma et al., 2021) also use a GCN encoder, which, according to the discussion presented in Section 2.2, may fail to distinguish nodes with similar local structure. In (Salehi & Davulcu, 2020), the authors propose to add a self-attention mechanism to the encoder, but their different goal is to reconstruct the features, and the method does not output a reconstruction of the graph structure using the latents.

**Algorithm unrolling for GRL.** Lastly, our method is based on the algorithm unrolling paradigm (Monga et al., 2021; Gregor & LeCun, 2010; Sprechmann et al., 2015). In their seminal paper, Gregor & LeCun (2010) made an important connection between iterative optimization methods and NNs; see also our brief review of this work in Section 2.3. Although there is an emerging interest in applying these ideas to inverse problems on graphs such as network topology inference (Shrivastava et al., 2020; Wasserman et al., 2023; Pu et al., 2021; Wasserman & Mateos, 2024), source localization (Ye & Mateos, 2022), or signal denoising (Nagahama et al., 2022; Chen et al., 2021), to the best of our knowledge this is the first adaptation to GNNs for computing node embeddings, given the graph structure.

## 6 Discussion and Future Work

We proposed LASE, a parametric model capable of learning to approximate the ASE of a given graph in an unsupervised fashion, even in cases where GNNs fail at this GRL task. The novel architecture is based on the unrolling of a GD algorithm that solves a matrix factorization formulation of the ASE problem. This unrolling process leads to a deep model, where each layer (or LASE block) consists of a superposition of GCN and GAT sub-modules. Interestingly, by default the GAT modules operate on a fully-connected graph, with attention coefficients equal to the inner product between the incident nodes' embeddings. We empirically demonstrated that few layers (in comparison to the iterations needed for GD to converge) of a trained LASE suffice to output embeddings whose accuracy is on par with the eigendecomposition-based ASE gold standard – even when the given graph has unobserved edges. Regarding scalability, we proposed an efficient training scheme using randomly sampled sub-graphs from the target (large) graph we wish to embed. This technique together with sparse attention mechanisms, allowed us to infer accurate spectral embeddings faster than highly-optimized eigensolvers in scientific computing libraries, or, other state-of-the-art iterative methods.

In addition to their adoption for clustering or other network-analytic tasks (e.g., visualization), spectral embeddings have been popularized as PEs. For this use case, spectral embeddings are typically precomputed and concatenated with any available node features before being fed to a GNN, whose output is used for downstream tasks such as node classification. We have verified how, as expected, the LASE PEs are as useful as those obtained via ASE, again with the added robustness benefit of tolerating a significant number

of unknown edges. Another one of our significant contributions is to demonstrate that LASE can also be used in an end-to-end fashion. Since LASE is a fully differentiable parametric function, we can embed it within a larger GRL pipeline and train the whole system using an augmented loss that combines prediction and reconstruction terms. The GNN's input is a concatenation of the output of LASE and the nodes' features, all in all resulting in a fully learnable PE architecture. Crucially, this eliminates the need for precomputing spectral PEs, achieving a prediction accuracy that is either superior to or competitive with methods requiring such computationally taxing offline step.

LASE opens up several research avenues, of which we highlight some of the most intriguing ones here. Firstly, the LASE architecture follows very closely the GD algorithm it was inspired from. For instance, we have not used any form of non-linearity in our basic model. Interleaving non-linear activations may increase the system's expressivity, generate sparsity in the GAT sub-module, or even mimic second-order optimization methods. Other modifications, such as skip connections mirroring momentum-based algorithms, may further reduce the number of layers required in LASE. We stress that is the algorithm unrolling principle that facilitates all these interesting potential links with advances in optimization theory and algorithms.

Secondly, there are several properties of LASE that we have empirically verified but deserve a theoretical study. For instance, the transferability of the learned weights in LASE that we used when training in smaller sub-graphs is based on the fact that (with the proper normalizations), the optimum of the reconstruction cost in the original and smaller graphs should coincide, in expectation. However, a more precise analysis of the impact on LASE's parameters of this method, or assessing the transferability between graphs with different spectral properties, is in order.

Furthermore, we have not considered directed graphs. In this case, the nodal representation includes two vectors, modeling the outgoing and incoming behavior of each node. Interestingly, these spectral embeddings can also be computed through a first-order iterative algorithm, although in this case it is necessary to consider certain Riemannian manifold constraints (Fiori et al., 2024). Taking an unrolling perspective on this method is clearly an interesting, albeit challenging, task that is part of our ongoing research agenda.

Finally, LASE (and its E2E variant) has broadened the applicability of ASE by showing that it can be an inductive, efficient alternative to the latter. In particular, we demonstrated its validity as a (learnable) PE and have shown promising results when compared to the PowerEmbed baseline. An interesting avenue of future research would be to further expand these comparisons to include other *non-spectral* PEs, that exhibit state-of-the-art performance in established downstream tasks such as node classification and link prediction.

### Acknowledgements

Work in this paper was supported in part by CSIC (I+D project 22520220100076UD).

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

## A  Appendix

### A.1  Graph convolutional filters: A step-by-step derivation

Graph convolutions are defined in terms of the so-called Graph Shift Operator $\mathbf{S} \in \mathbb{R}^{N \times N}$, a matrix representation of the graph $G(\mathcal{V}, \mathcal{E})$ that respects its sparsity; i.e. $S_{ij} \neq 0$ if $(j, i) \in \mathcal{E}$; see e.g., (Isufi et al., 2024). In the main body of the paper we used $\mathbf{S} = \mathbf{A} = \mathbf{A}^\top$ for illustrative purposes, but other possibilities, such as the Laplacian or their normalized counterparts are also viable choices. Assume that each node $i \in \mathcal{V}$ is endowed with a signal $(x_{\text{in}})_i \in \mathbb{R}$, then a first-order graph convolution is defined as

$$(x_{\text{out}})_i = (x_{\text{in}})_i h_0 + \sum_j S_{ij} (x_{\text{in}})_j h_1, \ i \in \mathcal{V},$$

where $h_0, h_1 \in \mathbb{R}$ are the filter coefficients. In words, the result of the convolution in a given node is a linear combination between its own signal and the sum of its neighbors' signals. This can be compactly written in terms of the matrix representation of the graph as

$$\mathbf{x}_{\text{out}} = \mathbf{x}_{\text{in}} h_0 + \mathbf{S} \mathbf{x}_{\text{in}} h_1$$

where $\mathbf{x}_{\text{in}}, \mathbf{x}_{\text{out}} \in \mathbb{R}^N$ are column vectors including all the nodes' signals. From this perspective, we may re-interpret $\mathbf{S} \mathbf{x}_{\text{in}}$ as a displaced or shifted version of $\mathbf{x}_{\text{in}}$. Arbitrary filters can be generated by repeatedly multiplying the input signal by $\mathbf{S}$ (i.e. repeatedly shifting the signal), resulting in the following general definition of a $K$-th order graph convolution:

$$\mathbf{x}_{\text{out}} = \sum_{k=0}^{K} \mathbf{S}^k \mathbf{x}_{\text{in}} h_k. \tag{17}$$

By virtue of the Cayley-Hamilton theorem, one has that $K \leq N$. This expression can be extended to include multi-dimensional signals. If the input and output signals have $F_{\text{in}}$ and $F_{\text{out}}$ features per node respectively, then $\mathbf{X}_{\text{in}} \in \mathbb{R}^{N \times F_{\text{in}}}$ and $\mathbf{X}_{\text{out}} \in \mathbb{R}^{N \times F_{\text{out}}}$, resulting in the final form of the graph convolution [cf. (2)]:

$$\mathbf{X}_{\text{out}} = \sum_{k=0}^{K} \mathbf{S}^k \mathbf{X}_{\text{in}} \mathbf{H}_k, \tag{18}$$

where $\mathbf{H}_k \in \mathbb{R}^{F_{\text{in}} \times F_{\text{out}}}$. To interpret (18), notice that the $F_{\text{out}}$ dimensions of the output signal correspond to a linear combination of the $F_{\text{in}}$ dimensions of the shifted input signal, where this last step does not include exchanged information between nodes.

### A.2  Graph convolution's output under white Gaussian noise and the symmetric SBM

Assume that in (17) we have that $\mathbf{x}_{\text{in}} \sim \mathcal{N}(\boldsymbol{\mu}, \boldsymbol{\Sigma})$ is white Gaussian noise, i.e., a standard multivariate Gaussian vector with mean $\boldsymbol{\mu} = \mathbf{0}$ and covariance $\boldsymbol{\Sigma} = \mathbf{I}_N$. Recalling the definition of GFT $\tilde{\mathbf{x}}_{\text{in}} = \mathbf{V}^\top \mathbf{x}_{\text{in}}$ and using basic properties of a Gaussian vector we have that $\tilde{\mathbf{x}}_{\text{in}}$ is also a standard Normal random vector. Furthermore, applying the GFT results discussed in Section 2.2 we conclude that $\mathbf{x}_{\text{out}}$ is a random combination of the eigenvectors $\mathbf{v}_i$, weighted by the filter's frequency response $\tilde{h}(\lambda_i)$, namely

$$\mathbf{x}_{\text{out}} = \sum_{i=1}^{N} \tilde{h}(\lambda_i) (\tilde{x}_{\text{in}})_i \mathbf{v}_i.$$

In particular, the $j$-th coordinate of $\mathbf{x}_{\text{out}}$ is a zero-mean Gaussian random variable with variance equal to $\sum_{i=1}^{N} \tilde{h}(\lambda_i)^2 (\mathbf{v}_i^2)_j$, where the squaring operation is to be conducted entry-wise. Note that if $\mathbf{v}_i^2$ is a constant vector for all $i$, then $\mathbf{x}_{\text{out}}$ is statistically identical for all its entries $j$. This is precisely the case for the symmetric SBM studied in Example 1. In particular, the two dominant eigenvectors are $\mathbf{v}_1 \approx 0.3 \times \mathbf{1}_N$ and $\mathbf{v}_2 \approx [0.3 \times \mathbf{1}_{N/2} \| -0.3 \times \mathbf{1}_{N/2}]$ with eigenvalues of $\lambda_1 \approx 0.3$ and $\lambda_2 \approx 0.2$ respectively. The other eigenvalues are orders of magnitude smaller.

| $\rho$ | No PE | ASE PE | LASE PE | LASE E2E | PowerEmbed RW-10 |
|---|---|---|---|---|---|
| 0 | $70.9 \pm 0.5$ | $71.1 \pm 0.5$ | $71.4 \pm 0.4$ | $\mathbf{71.8 \pm 0.5}$ | $71.2 \pm 0.4$ |
| 0.2 | $71.0 \pm 0.6$ | $71.5 \pm 0.6$ | $70.5 \pm 0.5$ | $\mathbf{71.8 \pm 0.3}$ | $71.5 \pm 0.5$ |
| 0.4 | $70.6 \pm 0.4$ | $71.0 \pm 0.6$ | $71.2 \pm 0.5$ | $\mathbf{71.3 \pm 0.4}$ | $71.1 \pm 0.4$ |
| 0.6 | $70.6 \pm 0.6$ | $70.7 \pm 0.6$ | $70.9 \pm 0.4$ | $\mathbf{71.0 \pm 0.3}$ | $70.5 \pm 0.4$ |
| 0.8 | $69.9 \pm 0.5$ | $70.0 \pm 0.5$ | $70.1 \pm 0.5$ | $\mathbf{70.3 \pm 0.3}$ | $70.1 \pm 0.4$ |

Table 5: Mean accuracy $\pm$ standard deviation over 10 data splits for the Twitch ES network with different proportions of missing edges $\rho$. The results with (*) trained using sparse attention WS with $r = 0.1$ and $p = 0.1$ instead of *full attention*. The results with (**) corresponds to PowerEmbed(RW)-2 taken from Huang et al. (2022) which uses the Random Walk matrix and 2 iterations.

| $\rho$ | No PE | ASE PE | LASE PE | LASE E2E | PowerEmbed RW-10 |
|---|---|---|---|---|---|
| 0 | $\mathbf{75.0 \pm 0.2}$ | $74.6 \pm 0.4$ | $74.6 \pm 0.3$ | $74.6 \pm 0.4$ | $73.2 \pm 0.5$ (**) |
| 0.2 | $73.7 \pm 0.5$ | $\mathbf{73.8 \pm 0.4}$ | $73.7 \pm 0.4$ | $73.0 \pm 0.5$ | $71.8 \pm 0.5$ |
| 0.4 | $\mathbf{73.0 \pm 0.5}$ | $72.7 \pm 0.3$ | $72.4 \pm 0.4$ | $72.7 \pm 0.7$ | $71.1 \pm 0.8$ |
| 0.6 | $72.3 \pm 0.6$ | $71.9 \pm 0.7$ | $72.3 \pm 0.5$ | $\mathbf{72.4 \pm 0.5}$ | $71.2 \pm 0.5$ |
| 0.8 | $71.8 \pm 0.4$ | $71.1 \pm 0.3$ | $\mathbf{71.9 \pm 0.7}$ | $71.0 \pm 0.2$ | $70.0 \pm 0.5$ |

Table 6: Mean accuracy $\pm$ standard deviation over 10 data splits for the Citeseer network with different proportions of missing edges $\rho$. The results with (*) trained using sparse attention WS with $r = 0.1$ and $p = 0.1$ instead of *full attention*. The results with (**) corresponds to PowerEmbed(RW)-2 taken from Huang et al. (2022) which uses the Random Walk matrix and 2 iterations.

In this context, consider the idea in (Kanatsoulis & Ribeiro, 2024). In order to avoid random outputs, they work with the output signal's variance instead, which is then fed to a GNN. Accordingly, for the symmetric SBM of Example 1 we obtain

$$\mathbb{E}\left[\mathbf{x}_{\text{out}}^2\right] = \sum_{i=1}^{N} \tilde{h}^2(\lambda_i)\mathbf{v}_i^2 \approx (\tilde{h}^2(0.3) + \tilde{h}^2(0.2)) \times 0.09 \times \mathbf{1}_N,$$

which is a constant, and thus cannot be used to discriminate between nodes. Note that we are neglecting the smaller eigenvalues, but under an SBM these are noise and should be ignored by any learning system.

### A.3 Further node classification results

As a complement to Section 4.2.2, results corresponding to the Twitch ES and Citeseer networks are reported in Tables 5 and 6, respectively. Just like for Amazon, all methods obtain approximately the same performance, and it is relatively independent of the proportion of unknown edges. This is indicative that the graph structure does not provide significant information, which we have verified by training a 2-layered fully connected NN with hidden dimension of 64, using only the node features as input. Node classification results for Twitch ES and Citeseer are $70.1 \pm 0.4$ and $72.5 \pm 0.3$, respectively.

### A.4 UN dataset details

Table A.4 shows the details of the UN dataset we have used to showcase embeddings under unknown edges and link prediction tasks. We include the years considered in Sections 4.1.4 and 4.2.1.

### A.5 LASE complexity

The complexity of each LASE layer consists of one graph convolution using the matrix $\mathbf{A}$ as the graph shift (or aggregation) operator, meaning a multiplication $\mathbf{AX}$; and one graph attention step that entails the

| Year | # Countries | # Resolutions | # Affirmatives | # Unknowns |
|------|------------|---------------|----------------|------------|
| 1960 | 100 | 54 | 2577 | 1360 |
| 1970 | 126 | 67 | 5182 | 2325 |
| 1980 | 153 | 103 | 11825 | 3371 |
| 1993 | 183 | 65 | 8177 | 3307 |
| 2004 | 191 | 90 | 12432 | 3629 |
| 2017 | 193 | 129 | 18086 | 5021 |

Table 7: Details on the UN dataset. For each year, we indicate the number of countries, resolutions, and how many votes were either affirmative or unknown.

multiplication $\mathbf{X}\mathbf{X}^\top\mathbf{X}$. If $\mathbf{A}$ is the adjacency matrix of a sparse graph with $|\mathcal{E}|$ edges, then the convolution can be computed with $O(|\mathcal{E}|d)$ operations. The attention term on the other hand, will require $O(N^2 d)$ operations in general, therefore being the leading complexity term. Now, when we sparsify the attention matrix with a mask $\mathbf{M}_{\text{att}}$ (Section 3.3), the complexity is reduced to $O(p_{\text{att}} N^2 d)$. The total inference cost for $L$ layers in LASE is then $O(L p_{\text{att}} N^2 d)$.

Regarding the number of learnable parameters in LASE, they amount to a total of $2 \times L \times d^2$ corresponding to $\mathcal{H} = \{\mathbf{H}_{1,l}, \mathbf{H}_{2,l}\}_{l=1}^{L}$. When using (L)ASE together with a GNN for a downstream task, necessarily the dimensionality of the input to the first layer increases by $d$ (compared to not using any PE). However, when using the precomputed ASE or the LASE output as a PE, the number of learnable parameters for the task corresponds to those of the GNN [as LASE is trained to solve (1), without considering the task at hand]. Only when using LASE E2E the learnable parameters add up to those in the GNN, increasing by $2 \times L \times d^2$.

