# OpenReview forum: "LASE: Learned Adjacency Spectral Embeddings"
_TMLR — Accepted by TMLR_

### Review · Reviewer_Qvgz · 2025-02-09

**Summary Of Contributions:**

The paper proposes a neural architecture to learn Adjacency Spectral Embeddings (ASEs), whose design is characterized as the unrolling of the Gradient Descent algorithm via two GNN layers: a GCN, and a GAT applied over a fully connected instance of the graph. Authors describe variations of the architecture considering performance, and design downstream architectures using LASE layers to learn self-supervised ASEs or tackle end-to-end node classification and prediction tasks.

**Audience:**

Yes

**Broader Impact Concerns:**

N/A.

**Claims And Evidence:**

No

**Requested Changes:**

See weaknesses and detailed review. In my view, given its current status, the paper requires (in order of importance):


### Critical changes:
#### These changes would provide theoretical and empirical support for the current claims.

* Additional experimental results to help show that LASE _is_ relevant: comparisons with other PE methods, fairly considering model capacity (i.e. parameter counts) and depth, in common node / edge / graph-level tasks (e.g. datasets from Dwivedi et al., 2022, or the Open Graph Benchmark).
* A comparison of the time and memory complexity of LASE versus existing approaches, together with whether existing methods are transductive (i.e. only apply to known nodes / edges / graphs), or inductive.
* Improved empirical comparisons with existing approaches, at the very least containing approaches that are currently cited (e.g. Huang et al., 2022; Lim et al, 2023.), and ideally at least one example of other PE methods (e.g. sub-graph or substructure based).

### Desirable improvements to strengthen the work:
#### These changes would improve positioning the work against existing approaches, provide a more thorough picture of the merits of LASE, how it has been evaluated, and whether the experimental comparisons are fair.

* A more systematic analysis of the empirical results regarding model performance and costs: what are the training and inference time / memory cost for each of the models in Tables 1, 2 and 3?
* Better positioning with the state of the art: why is being able to efficiently approximate ASEs a relevant problem? how does it compare with existing approaches? which benefits does it bring (interpretability, performance, reduced computation or memory costs, ability to model higher-order signals, expressivity according to some test like Weisfeiler-Lehman, ...)?
* Additional dataset details, as an appendix describing the datasets, number of graphs, avg. number of nodes, edges, number of features, classes, and objective.
* Additional hyper-parameter details, including all explored hyper-parameters, how hyper-parameter search was conducted, if at all, and the number of hyper-parameters tried on each experiment for the models being evaluated.

**Strengths And Weaknesses:**

This section summarizes strengths and weaknesses first, and then provides a detailed review of the paper including comments and questions.

### Strengths
* The paper thoroughly describes the derivation from the ASE formulation to their proposed Learned ASE (LASE) embeddings produced via algorithm unrolling.
* The writing is clear and accessible, with a notation that can be easily parsed and understood.
* The relationship between ASE, LASE, algorithm unrolling, and their derivation shows some promising direction.
* The proposed LASE method is inductive, can be trained in an end-to-end fashion, and the authors show certain cases in which it is robust to novel inputs and limited distribution shifts.

### Weaknesses

Note that major concerns are described in top bullets, with details nested.

* The derivation of how LASE is designed does not provide any theoretical or practical insights — although it seems correct, it is not clear why it is particularly relevant on either formal grounds or specific applications.
  * No formal assessment about the time / memory complexity of training and inference for LASE, and direct complexity comparisons with existing methods.
  * Missing motivation for why ASE approximation is beneficial versus other Positional Encoding (PE) approaches.
  * No theoretical analysis of model capacity / expressivity compared to existing spectral approaches.
  * Unclear advantages over simpler positional encodings or non-spectral alternatives.
* Several aspects of the paper are underspecified or poorly motivated:
  * The application of sparse attention as a mask is not clear, particularly in how the attention mask is constructed and why.
  * Upon reviewing the code, it seems that the sparse attention mask is just a set of randomly generated edges from a generative model (like Erdos-Renyi or Watts-Strogatz), with no relationship to the actual connectivity of the graph. This seems to me poorly motivated, specially as it seems to be a random connectivity structure overlaid over the graph with no theoretical basis or guarantees.
  * Details about the training procedure (number of iterations, batching if applicable, other hyper-parameters) are missing.
* The experimental evaluation has significant limitations:
  * Dataset and task details are underspecified.
    * For example, the authors provide a small-scale evaluation on link prediction task, with underspecified dataset details (number of graphs, average $|V|$, $|E|$, degree, etc.)
    * An appendix providing dataset and task details would help clarify the data the authors rely on to make their claims.
  * The analysis of robustness only considers a naïve form of distribution shift, and does not consider perturbations of the graph structure.
  * Empirical timing results are unclear, in particular whether the time is relative to obtaining a comparable result, or at least the results described in any of the task-specific experiments.
  * Most reported improvements seem to be statistically insignificant.
    * All results in Table 3 seem to be statistically indistinguishable from the first column:
    * In Table 1, statistical significance cannot be assessed (there's no stddev).
    * In Table 2, stat. sig. improvements only show for $\rho > 0$, which is expected given the inductive setup.
  * Limited comparison with state-of-the-art positional encodings (see detailed review for related PEs, at least some of which should be compared against).
    * Although the paper presents a spectral method, its position as a PE means the state-of-the-art is broader than just ASE as a set of features: the authors need to compare with other PEs that offer inductive, learnable representations and assess whether the costs, benefits, and model complexity complexity indicate LASE is preferrable.
  * Key architectural choices (e.g. LASE depth, attention masks) are not validated through data (e.g. ablation studies for predictive performance and computational costs).
  * The comparisons seem unfair regarding model capacity: they seem to feature a GNN baseline ("No PE") compared against the same GNN that is fed an input from $L$ LASE layers.
    * There is significantly more depth in the second case, particularly considering that LASE layers can be learning weights over both local connectivity and the fully-connected graph.
    * There are no comparisons on the basis of training / inference cost for both, nor accounting for the # of learnable parameters, as is typically done when comparing GNN architectures.

### Detailed review

* Page 2: What is the problem with these precomputed PEs? If time / memory cost, it would be helpful to be explicit about their limitations. There exist inductive, non-Spectral PEs that have acceptable time complexity (e.g. counting or encoding structures and subgraphs). If the claim is that LASE is less costly than Spectral PEs, the authors should still compare with non-spectral PEs to motivate why LASE is desirable (due to e.g. interpretability, the ability to capture global structures, etc.).
* Page 5: Besides random signals to identify each node, other approaches encode (local) structural patterns so that nodes are identified by the (usually permutation-equivariant) structures surrounding them (Zhao et al., 2021, Bouritsas et al., 2023;  Alvarez-Gonzalez et al., 2024). These methods remain inductive, are deterministic, and have been shown to be more expressive than vanilla GNNs with moderate additional costs.
* Page 8: GATs are defined in terms of  messages through node neighbourhoods with attention. Framing the $X_l X_l^T$ term as a GAT on a fully connected graph seems wrong, considering both the message-passing aspect and the softmax term are omitted. It is a node similarity matrix where each entry is the dot product of node pairs.
* Page 9: The authors claim “The unrolling offers an explicit handle on complexity leading to faster (post training) inference times” but do not explicitly analyze the time complexity of their proposed approach. This echoes the introduction: “Learned ASE (LASE), which is interpretable, parameter efficient, robust to inputs with unobserved edges, and offers controllable complexity during inference.”).
  * What is the worst-time cost of training L LASE layers in terms of $|V|$, $|E|$, $L$ and $d$?
  * Are the LASE layers simply trained via backprop with SGD on mini-batches?
  * Similarly, what is the cost of inference?
  * Authors imply the forward cost of the GCN and GAT layers and can be stopped at a chosen depth to provide a balance between inference cost and output quality — but the time complexity alone vs. a GNN or a Graph Transformer is not discussed. After going through the paper, as far as I can see, they also do not analyse the impact of 'stopping early' during inference.
* Page 10: How are the proposed attention masks computed? Since they determine a subset of the fully-connected set of edges, they should also affect whether $\mathbf{X}_{l+1}$  is a good approximator of the vanilla ASE definition. The authors mention the mask ‘encodes the topology of the sparse graphs’ — how? which topology? based on substructures, subgraphs, or something else? This seems like a major architectural choice that is underspecified in the manuscript.
  * Reading through the paper, it seems that in Page 16. they define the mask by sampling edges via Erdos-Renyi or Watts-Strogatz models — but these methods are generative, so it is not immediately clear how they would apply to an existing graph.
  * Reading through the code (under `./training/train_glase_e2e_classifier.py`), it seems the authors either generate a fully-connected graph or a random Watts-Strogatz graph with a matching number of nodes, but it's unclear how that would be relevant for a given arbitrary input graph and the choice of 'overlaying' a randomly generated graph is poorly motivated. I am refering to the following code snippet:
```python
if att_mask == 'FULL':
    edge_index_2 = torch.ones([num_nodes,num_nodes],).nonzero().t().contiguous().to(device)
else:
    edge_index_2 =  from_networkx(watts_strogatz_graph(num_nodes, 700, 0.1, seed=None)).edge_index.to(device)
```
* Page 12: Given the objective as stated in Eq. 4., what would be the outcome if the LASE layers were not structure aware, i.e. they learned a mapping $f: \mathbf{X} \in \mathbb{R}^d \mapsto \mathbb{R}^{\hat{d}}$ such that $f(x_i) \cdot f(x_j) \approx A_{i,j}$ via an MLP? It is clear this would be a transductive, rather than inductive setting, as we’d be learning to map from each graph’s node feature distribution onto their nodal adjacency spectral embeddings. However, this would also provide baselines for the delta in performance between inductive and transductive cases, which currently are not clear, especially in terms of the relevance of Figure 7.
* Page 13: Is each sample seen once, and trained via a batch-size-of-1 SGD? What is the actual training process for LASE? This is a key point that is not clear.
* Page 14:
  * Minor: is $\Pi_2$ renormalized? It must be in order to remain a valid connection probability matrix where rows / columns add up to 1.
  * How well do these results hold? E.g. if a LASE model is trained in a range $\delta \in (-0.01, 0.01)$, does it generalize to $(-0.1, 0.1)$? How about $\pm 0.2, 0.4, …$? The SBM is a rather simple model, so assessing robustness from them is not clear, and it doesn’t inform us about the upper bound of what distribution shifts it tolerates. How about the real world graphs: does a LASE model trained on Cora remain valid on the CiteSeer or Amazon datasets? How about perturbations from a given graph (e.g. via node / edge removal)?
* Page 15: Which of the four attention variants are used in the paper for reporting other empirical results? What is the time and memory consumption between the proposed methods? Which result shows the ASE baseline from Graspologic? Are the wall times computed with respect to reaching comparable performance? On which dataset / task?
* Page 16: (Figure 1) What does it mean to perform competitively, in particular compared against the standard ASE that should serve as the baseline? Additionally, are the reported differences between losses statistically significant?
* Page 18:
  * This link prediction task seems rather small: how many graphs are contained in the dataset? What’s their average number of nodes and edges? Are the differences between “No PE”, “ASE PE” and “LASE PE” statistically significant?
  * Link prediction is a standard task in GRL with datasets going up to traditional node embeddings like e.g. node2vec (Grover et al., 2016). The authors should evaluate their method in a larger benchmark with more methods compared against — to assess why / whether ASE and LASE are relevant.
  * How were the hyper-parameters chosen ($L = 10$, $d = 4$)? Since $L = 10$ implies 10 LASE layers, each of which are potentially fully connected in the GAT, the comparison with a standalone GNN is hardly fair, particularly in the end-to-end case. Is the GNN architecture only a GCN? What if you use a GAT? What if the GNN without LASE has more stacked layers? An appendix describing how hyper-parameters were chosen, and how hyper-parameter search was performed, if any, would be valuable. It would also assess whether the experiments are fair and comparable, inasmuch when more experiments are performed for method A than method B, it is more likely that method A will hit a 'lucky' result.
* Page 20:
  * (Table 2): Same questions as above: what is the model, how are hyper-parameters chosen, why are these hyper-parameters comparable? Additionally, at least for the $\rho = 0$ case in Table 2, results are statistically insignificant.
  * (Table 3): All results between LASE E2E (highlighted) and the No PE columns seem to be statistically insignificant (. Why is there no comparison with GNNs (e.g. GCN, GAT, and sub-graph / positional GNNs and PEs like GNN-As-Kernel, Shortest Path Neural Networks, Equivariant Subgraph Aggregation Networks, Edge-Level Ego-Network Encodings, Subgraph Union Networks, …)?
* On related work: Are there time / memory complexity differences with PowerEmbed? If not, it is hard to assess why LASE would be preferrable, as the experiments cover only two comparisons.
* Page 21: Authors cite Puny et al. 2020, which seems relevant (and more applicable than GATs) but do not relate their results to their method. The expressive power of matrix operations has been studied in Matlang (Brijder et al, 2019) and has found applications to GNN design in GNNML1 / GNNML3 (Balcilar et al. 2021). There have also been direct connections between the expressive power of GNNs with spectral information (Wang et al., 2022) that provide connections with other related works. For instance (Chien et al., 2021), proposed a GNN architecture to that is provably more powerful by learning a Generalized PageRank. In these works, which in my view are related to manuscrupt inasmuch as they construct or learn signals from graph spectra, the theoretical analyses and experimental approaches may help better position the presented work.

### References

* Adaptive Universal Generalized PageRank Graph Neural Network. Chien et al., in International Conference on Learning Representations, 2021.
* Benchmarking Graph Neural Networks. Dwivedi et al., in Journal of Machine Learning Research, 2023.
* Breaking the Limits of Message Passing Graph Neural Networks. Balcilar et al., in International Conference on Machine Learning, 2021.
* From Stars to Subgraphs: Uplifting Any GNN with Local Structure Awareness. Zhao et al., in International Conference on Learning Representations, 2022.
* How Powerful are Spectral Graph Neural Networks. Wang et al., in International Conference on Machine Learning, 2022.
* Improving Graph Neural Network Expressivity via Subgraph Isomorphism Counting. Bouritsas et al., in IEEE Transactions on Pattern Analysis and Machine Intelligence. 2022.
* Improving Subgraph-GNNs via Edge-Level Ego-Network Encodings. Alvarez-Gonzalez et al., in Transactions of Machine Learning Research, 2024.
* node2vec: Scalable Feature Learning for Networks. Grover et al., in International Conference on Knowledge Discovery and Data Mining (KDD), 2016.
* On the Expressive Power of Query Languages for Matrices. Brijder et al., in ACM Trans. Database Syst.  2019.

---

> ### Author Response · Authors · 2025-04-16
> **Authors’ response to Reviewer Qvgz (part 1 of 4)**
>
> Thanks for your time and effort in reviewing our paper. We are glad to hear you found our method to be promising and its derivation thorough, the presentation clear and accessible,  as well as recognizing LASE’s attractive features. We appreciate your detailed assessment of our work and all the valuable suggestions for improvement. Accordingly, we have implemented revisions to address the major issues raised. Below, we provide responses to your comments and requested changes. We strive to improve our paper and will be happy to continue the discussion if any outstanding issues remain.
>
> Comment:
>
> **Theoretical or practical insights on the derivation of LASE**
>
> We believe that LASE provides an important fundamental insight: a simple but well motivated modification to a basic GNN is sufficient to express the spectral decomposition of the underlying graph, and the resulting weights are transferable to other graphs. The architectural construction process follows a *principled* approach that relies on algorithm unrolling of GD iterations, which have well documented merits when it comes to solving the (refined) matrix factorization formulation (1) of ASE. We acknowledge that we do not formally establish this transferability property of the proposed LASE architecture. But such theoretical study is well beyond the scope of this paper and, respectfully, should not take away from its significant contributions.  Instead, we have *empirically* shown how these learnt weights are fairly robust to shifts in the underlying distribution. Unlike PowerEmbed, LASE operates purely through message passing and eliminates the need for a normalization step, which involves aggregating all node embeddings and performing a matrix inversion -- a potential bottleneck as $d$ increases. Moreover, LASE can form these spectral embeddings even in scenarios where edge information is missing, typical in link prediction tasks and other settings we explore throughout our comprehensive experimental evaluation.
>
> From a practical standpoint, reinterpreting the GD method as a modified GNN via the principle of algorithm unrolling offers two distinctive advantages. First, it enables learning parameters that markedly reduce the number of required iterations (or layers) compared to GD, while still attaining competitive nodal embedding accuracies. Second, it allows for seamless integration with specialized software libraries, such as PyG. All in all, **LASE yields node embeddings that are robust to missing data as well as minor distribution shifts, at a fraction of the cost of prior art and without sacrificing approximation accuracy.**
>
> We stress that computing the ASE is a fundamental problem in its own right, independently of its usage as a positional encoding (PE). On top of unsupervised learning of ASEs that are useful e.g., in graph visualization, clustering, hypothesis testing, or change-point detection, we show LASE may also be used in an end-to-end (E2E) fashion for other supervised tasks such as link prediction of node classification. Our results show LASE E2E is competitive with other methods, in particular when pre-computing the PEs, effectively bypassing this potentially expensive step. In any case, as we now explicitly spell out in the revised Introduction **our objective in this work is not to propose a new state-of-the-art PE, but rather to broaden the applicability of ASE – whose merits for statistical inference have been well documented.** Please see also our response to the LASE motivation comment by Reviewer Rta7.
>
> Comments along these lines have been included in the revised Introduction, which we believe now better conveys the significance of our work. We have also conducted additional experiments, including one where we study the quality and robustness of the obtained embeddings in the UN dataset, where missing relational data are prominent (and ASE is not directly applicable unless one arbitrarily sets unobserved edges to zero); please check Section 4.1.4. Thanks again for your valuable feedback.
>
> **Clarifications on the experimental procedure**
>
> Thanks for bringing up these clarification requests. With regards to the usage of the attention mask in the GAT sub-module, the unrolled GD layers in (7) call for a fully-connected graph. Note that this fully-connected graph is unrelated to $\mathbf{A}$. Inspired by transformers and the sparse mechanism typically used in that context, we have explored the use of a sparse attention mask $\mathbf{M}\text{att}$, where nodes only exchange information with a fraction of the other nodes. For instance, in the ER case, each node would perform a message passing round with randomly selected nodes. Again, we reiterate that this sparse structure induced by $\mathbf{M}\text{att}$ is unrelated to graph topology encoded in $\mathbf{A}$. We have revised Section 3.3 to clarify these points; please check the discussion following (10).

---

> ### Author Response · Authors · 2025-04-16
> **Authors’ response to Reviewer Qvgz (part 2 of 4)**
>
> Regarding the LASE training procedure, the training set comprises pairs each consisting of a graph $\mathbf{A}^{(i)}$ and a noise input $\mathbf{X}_0^{(i)}$. For instance, in Example 2 we used a training set of $T=100$ samples drawn from the same random graph model (a symmetric SBM) and independent noise inputs. Training is performed through mini-batches from this training set, using backpropagation and SGD to minimize the empirical risk in (4). In all other experiments we proceeded similarly. Moreover, whe we use graph sub-sampling, each training point contains a randomly sampled subgraph from the larger target graph to be embedded, whereas in the E2E scheme the training set is augmented with the corresponding labels. We have included a more detailed description of the training procedure in the opening of Section 4. Following your suggestion, we make special reference to the chosen attention mask (capturing all-to-all interactions unless otherwise stated).
>
> **Further experiments to evaluate LASE’s robustness**
>
> Following your suggestion, we have expanded the experimental evaluation in Section 4 to illustrate LASE’s robustness to changes in the underlying graphs’ distribution. We agree this is an important feature, and we have considered two new test cases. First and relying on synthetic data, we have perturbed the connection probability matrix of the SBM separately for each entry (whereas before the same value was added to all entries). Note that the connection probability matrix for an SBM only requires that all its entries are between 0 and 1, since they indicate the connection probability between nodes of each class; please check the revised Section 4.1.2 for these new results. Additionally, we have perturbed one of the real-world networks (the Cora dataset), removing or adding links with a certain probability. These results are reported in the revised Section 4.1.4.
>
> Our findings are consistent. LASE is fairly robust to small graph perturbations, and as the model drifts significantly from the nominal one seen during training, performance degrades. However, including data points (i.e., graphs) drawn from the modified distributions in the training set enables LASE to recover accurate embeddings; see Figures 7 and 10 in the revised manuscript. The conclusion is that the architecture is expressive enough to accommodate (and learn from) several network distributions.
>
> Finally, we have also included an experimental study to show that LASE is robust to the chosen model hyperparameters. The proposed architecture only has two hyperparameters: the number of layers $L$ and the embedding dimension $d$. Results for the Cora dataset are reported in Figure 9 of the revised manuscript. We find that – except for limited and  somewhat pathological cases (such as excessively large dimension $d$ and small $L$ – the resulting reconstruction loss is fairly insensitive to the choice of hyperparameters.
>
> **Improved empirical comparisons with related approaches**
>
> Thanks for this valuable suggestion. We have now included expanded comparisons with PowerEmbed in all end-to-end tasks, including its evaluation in link-prediction for the UN dataset and different missing data fractions \(\rho\) for the node-prediction case in Cora, Citeseer, Twitch ES and Amazon. Please check the updated tables and accompanying discussions in Section 4.2 and Appendix A.3 of the revised manuscript.
>
> **Time and memory complexity comparisons**
>
> This point is well taken. The complexity of each LASE layer consists of one graph convolution using the matrix $\mathbf{A}$ as the graph shift (or aggregation) operator, meaning a multiplication $\mathbf{A}\mathbf{X}$; and one graph attention step that entails the multiplication $\mathbf{X}\mathbf{X}^\top \mathbf{X}$. If $\mathbf{A}$ is the adjacency matrix of a sparse graph with $|\mathcal{E}|$ edges, then the convolution can be computed with $O(|\mathcal{E}|d)$ operations. The attention step on the other hand, will require $O(N^2d)$ operations in general, therefore being the leading complexity term. Now, when we sparsify the attention matrix with a mask, as proposed in Section 3.3, the complexity is reduced to $O(p_{att}N^2d)$. The total inference cost for $L$ layers in LASE is then $O(Lp_{att}N^2d)$. Comments along these lines have been included in the new Appendix A.5.

---

> ### Author Response · Authors · 2025-04-16
> **Authors’ response to Reviewer Qvgz (part 3 of 4)**
>
> When it comes to comparing the time complexity of LASE versus a GNN or a Graph Transformer, since the GCN and GAT layers are actually part of our architecture the complexity will be roughly equivalent – provided that the same number of parameters are chosen for both alternatives. Our claims on controllable complexity during inference pertained to the natural comparison against the iterative GD solver (which serves as the blueprint of the LASE architecture via unrolling). The latter typically requires hundreds or thousands of iterations to converge to the minimizer of (1), while LASE inference entails a forward pass through a handful (or more if so desired) neural network layers; see also Figure 8.
>
> **Refined analysis of the empirical results pertaining to model performance and cost**
>
> On the timing experiments reported in Section 4.1.3, they pertain specifically to LASE and we compare it to Graspologic, the gold standard for computing the ASE. Since the Graspologic ASE module is based on a truncated SVD decomposition of the adjacency matrix, we have labeled it as “SVD Truncate”’ in Figure 8. To conduct a fair comparison, LASE was configured so that when using the fully-connected mask the resulting total loss is on par with the one obtained by Graspologic, and within 5% when using the attention mask. We have underscored this criteria in the revised manuscript.
>
> Regarding the standard deviation of the results reported in Table 1 (i.e., the total loss for ASE, GD, and LASE when applied to four real-world networks), since they pertain to embeddings of individual graphs the variability is expected to be quite low. The ASE exhibits no variability, since it is a deterministic method. GD and LASE are randomly initialized, the GD method is known to be robust to this initialization and generally converges to the optimum in symmetric low-rank matrix factorization problems. Following your suggestion, we have nevertheless included the standard errors in all cases and commented on this aspect in the revised Section 4.1.4.
>
> Finally, moving on to the results in Table 4 (i.e., node classification in Amazon Photo), we find they are statistically indistinguishable for all methods since the actual structure of the network is uninformative for this task. As we discuss in the paper, a simple fully-connected neural network fed the nodes’ features as input is capable of producing the same results; see the closing paragraph of Section 4.
>
> **Positioning in context of state-of-the-art approaches**
>
> This point is well taken and we agree that the motivation behind LASE over the eigendecomposition-based ASE could have been better articulated (especially in the Introduction). We argue that spectral decompositions of the adjacency matrix may face three main limitations:
>
> 1. The eigendecomposition requires full-availability of the adjacency matrix, and it cannot accommodate missing edge data due to e.g., sampling, memory, or privacy constraints.
> 2. High computational complexity, especially in inductive settings whereby one would like to embed multiple moderately large graphs sampled from a common distribution.
> 3. Modifications to the graph require recomputing an eigendecomposition from scratch to obtain the new embeddings (except for specific, e.g., rank-one perturbations).
>
> As we clearly spell out in the revised Introduction, we thus contend that a new architecture is needed when it comes to learning to approximate ASEs. Our objective here is not to propose a new state-of-the-art PE, but rather to broaden the applicability of ASE – whose merits for statistical inference have been well documented. In particular, the ASE has been a workhorse graph representation learning technique due to its interpretability and statistical guarantees. The interpretability, which is crucial for a host of data science and machine learning applications, can be traced to the spectral underpinnings of this embedding method. Since it has been extensively reported in the literature (see for instance the original paper by Scheinerman and Tucker or the survey by Athreya et. al), we did not double up on this aspect in the manuscript. From a theoretical point of view, the ASE enjoys strong consistency and asymptotic normality results, and therefore it remains as a strong node embedding technique in the literature.

---

> ### Author Response · Authors · 2025-04-16
> **Authors’ response to Reviewer Qvgz (part 4 of 4)**
>
> Limitation 1 is addressed by the low-rank matrix decomposition formulation (1) and the corresponding GD solver in which LASE is based on. By including the mask $\mathbf{M}$, we render LASE applicable to more general graph representation learning settings. Running GD iterations until convergence incurs a computational cost that typically exceeds (highly optimized) eigendecomposition routines that can be used for ASE. However, and as shown in Fig. 8, LASE is an order of magnitude faster than the (more efficient) Block Coordinate Descent variant of GD. Furthermore, the inclusion of attention masks (see Section 3.3) results in shorter computation times than even those optimized eigendecomposition routines in Graspologic.
>
> Finally, since the LASE architecture combines a GCN and a GAT, it may be applied to any graph. Naturally, the accuracy of the obtained embeddings will depend on the underlying distribution. As we empirically show in Section 4.1.2, LASE offers some valuable degree of robustness in this regard. We fruitfully leverage this attractive property to decrease training times (by training on smaller sampled subgraphs of the target large graph to be embedded), thus addressing Limitation 3, as inference under modest changes still yields accurate results. All in all, our experiments provide evidence to support LASE yields node embeddings that are robust to missing data as well as minor distribution shifts, at a fraction of the cost of prior art (addressing Limitation 2) and without sacrificing approximation accuracy.
>
> **Additional dataset details**
>
> Following your suggestion, we have added a new Appendix A.4 with a table including formerly missing details about the UN dataset for each year we studied in our experiments. We report the number of countries, resolutions, how many votes were affirmative and how many were unknown. For all the other datasets, we have also included the number of edges, which was missing in the original submission; please check the revised opening paragraph in Section 4.1.4.
>
> Thanks again for your review.

---

> > ### Comment · Reviewer_Qvgz · 2025-04-29
> > **Thank you for addressing the feedback (1/2).**
> >
> > I want to thank the authors for their thorough responses and addressing the bulk of my comments. I believe the claims in the manuscript are better grounded with the presented results and the exposition has become more clear. Please find comments on a section-by-section basis below.
> >
> > > Theoretical or practical insights on the derivation of LASE, positioning in context of state-of-the-art approaches
> >
> > Thank you for better positioning the paper with respect to the state-of-the-art and providing a clearer outline of the objectives and rationale for the work. Your updates in the manuscript address concerns about comparing LASE to non-spectral nodal PEs, which now are presented as a means of validating that LASE can be an inductive, efficient alternative to ASE, rather than as a use-case in which LASE would perform on-par or better than the state-of-the-art.
> >
> > I would nevertheless argue that for completeness, it would be valuable to understand how LASE fares against recent PE approaches in known applications (like e.g. node-level classification tasks, link prediction). This would provide a clear rationale in downstream tasks for LASE besides its connection to ASE as a learnable, inductive variant. That said, I understand this request would be best taken as future work.
> >
> > > Clarifications on the experimental procedure, refined analysis of the empirical results pertaining to model performance and cost, additional dataset details
> >
> > Thank you for providing additional details on your experimental design and datasets. I believe that the configuration of experiments, particularly as it pertains to masks and the training procedure are easier to understand.
> >
> > Regarding sparse attention masks, the comment that the results are all within 5% of the full-attention loss is helpful. One caveat is that I still cannot see a rationale for the choices of masks: is there any reason for using an edge-mask that is produced by a graph generative model like Watts-Strogatz? I can understand that _by analogy_, a mask where each node has a uniformly-random probability $p$ of being connected to another maps to Erdös-Renyi (though I believe referencing a generative model is less clear than describing precisely that mask).
> >
> > What is the rationale for masking features with a randomly generated small-world graph? The same question applies for the regular lattice mentioned when describing BigBird attention, which is reasonable in text to model a sliding window of tokens, but in a graph with arbitrary (and permutable) node assignments. It would be helpful to describe the reasoning for this experimental design besides the empirical results to understand whether the same approach would be applicable to other graph domains.
> >
> > > Further experiments to evaluate LASE’s robustness, improved empirical comparisons with related approaches
> >
> > Thank you for extending the empirical results and digging deeper into the robustness of LASE. The new results convincingly show that LASE can be used in inductive scenarios with some guarantees re: performance degradation.
> >
> > One caveat is that model capacity (i.e. total count of parameters of the model) is still not described anywhere as far as I can tell. This means that when comparing the different approaches in Tables 3 and 4, it is not clear to which extent the total number of parameters differs between entries (and whether the comparison is fair). This is important to understand how much additional capacity introducing LASE versus the vanilla model without any PE, and also an existing method like PowerEmbed as a baseline. My expectation is that:
> >
> > * No PE < ASE PE < LASE PE == LASE E2E
> > * ASE PE only affects the size of the input in the first layer
> > * LASE PE and LASE E2E affects the size of the input in the first layer, plus the capacity of the LASE model
> >
> > Is this the case? I would recommend noting the actual model capacity in an appendix, where the increase in the number of parameters caused by incorporating LASE can be compared with the vanilla, No PE model.

---

> > ### Comment · Reviewer_Qvgz · 2025-04-29
> > **Thank you for addressing the feedback (2/2).**
> >
> > > Positioning in context of state-of-the-art approaches
> >
> > Thank you for extending the discussion and drawing more relationships to the state of the art.
> >
> > One important caveat to your comment ("LASE yields node embeddings that are robust to missing data as well as minor distribution shifts, at a fraction of the cost of prior art"), is that this is applicable when comparing LASE to purely spectral embeddings. However, it is not generally the case with inductive node embeddings, like the sub-graph or sub-structure methods mentioned in the review (2nd bullet of the detailed review), which have moderate time and memory cost.
> >
> > As these methods focus on sub-graph structures, they also address a weakness of methods leveraging random signals in inductive settings, which the latest version of the manuscript mentions as a point of comparison with prior art "[…] While simple, these schemes seriously limit the transferability of the learned architecture to other graphs, and may lead to functions that are not equivariant to node permutations"). I would recommend a more balanced characterization here, which in my view is that spectral methods can capture global properties while these sub-graph and structural methods often focus on local properties of graphs.
> >
> > > Time and memory complexity comparisons
> >
> > Thank you for expanding the manuscript with the additional analysis on time complexity. As noted in the review, I believe this section highlights how the current framing of LASE as a combination of a GCN and a GAT is misleading. GATs are MP-GNNs with comparable time complexity per attention head to GCNs (see Section 2.2 of the Velickovic et al, 2018 GAT paper). As other MP-GNNs, the structure that GATs process is given by the input network --- it is due to this that popular libraries will expect both a feature matrix and a representation of the network, e.g. node tuples encoding edges through `edge_index` in PyG.
> >
> > In the manuscript, GATs are used in a non-standard way where graph connectivity is ignored (as you compute all pairwise similarities, conditioned with a mask) and the attention mechanism is dropped. Although I understand the intent of connecting the proposed method to existing architectures, I believe the current framing needlessly complicates the contribution of the paper. I would recommend _not_ using the GAT reference as more than an analogy, though given how central it is to how the paper is written, I understand this is a major change. As a middle ground, I would encourage you to aid the reader, using `GNN(X, A)` notation in e.g. Figure 3 and other architecture diagrams, and to explicitly note that the attention component being used in LASE is not a straightforward GAT (rather than a comment in passing in Page 8 re: the activation), highlighting the differences.

---

> > > ### Author Response · Authors · 2025-04-30
> > > **Thanks**
> > >
> > > Thanks for checking our responses and the revised manuscript. We are glad to hear you found the claims in the manuscript are better grounded with the presented results and the exposition has become more clear.
> > >
> > > Your follow-up feedback and valuable suggestions are much appreciated. We are currently working on incorporating the suggested changes and will soon upload a revision to the paper (if we still can, else in the camera-ready version) along with a point-by-point response to your comments/caveats that we will post in this forum.
> > >
> > > Thanks again for engaging in the discussion as well as for your constructive comments that have led to a much improved and clear paper.

---

> > > > ### Author Response · Authors · 2025-05-06
> > > > **Authors’ response to Reviewer Qvgz’s follow-up comments (part 1 of 2)**
> > > >
> > > > We want to thank you again for your insightful feedback and valuable suggestions. The time and effort you have spent reviewing our manuscript is much appreciated. Point-by-point responses to your follow-up comments follow. We have uploaded a re-revised version of the manuscript, where the final minor comments implemented have been color coded blue.
> > > >
> > > > **LASE against recent PE approaches**
> > > >
> > > > Thanks for recognizing the improvements made in the revised manuscript, in terms of clarifying the objectives and scope of our work, as well as the positioning of LASE relative to existing spectral PE methods. We agree that a comprehensive comparison with recent non-spectral PE approaches on established downstream tasks (such as node classification and link prediction) would provide additional insights into the practical advantages and limitations of LASE. While such an evaluation falls outside the current scope of our (already fully-packed) manuscript and it should not take away from the paper’s contributions, we consider it a valuable direction and we are eager to pursue it as part of our future research agenda. Following your suggestion, we have now included this as an extra item in the future and related work discussion; please check the closing paragraph in Section 6 of the re-revised manuscript.
> > > >
> > > > **Rationale for masking features with randomly generated small-world graphs**
> > > >
> > > > In the fully connected graph used in the GAT sub-module all nodes exchange information with all other nodes in a single message passing round. And we argue this performance bottleneck may be alleviated by using a sparse graph instead of a fully-connected one. For the information to percolate efficiently among nodes using fewer number of message passing rounds, the graph should have a small diameter. This is the rationale behind choosing randomly generated small-world graphs and the reason why the Watts-Strogatz or the Big-Bird (BB) masks works better than an Erdos-Renyi for a given sparsity level. Please note that the regular lattice of the BB in this case is simply the starting point of the Watts-Strogatz model. Instead of randomly re-wiring edges, edges are randomly added in the BB case. An interesting non-random alternative would be for instance using expander graphs, as proposed in
> > > >
> > > > Deac, A., Lackenby, M. &amp; Veličković, P., ``Expander Graph Propagation,’’ *Proc. First Learning on Graphs Conference*, 198:38:1-38:18, 2022,
> > > >
> > > > where the graph is completely substituted by a small-diameter one in some of the layers. This is a promising idea, but the construction of such expander graphs (in particular tailored for our application) is non-trivial, and therefore left as future work.
> > > >
> > > >  **Total count of model parameters**
> > > >
> > > > As stated in the revised manuscript, the total number of LASE parameters is $2\times L\times d^2$; see `Properties of the LASE architecture'. When using a GNN with no PE or with a pre-computed one (as ASE or LASE), the only learneable parameters are those of the GNN. In the case of the pre-computed PEs, these are concatenated to the input, thus, as you mention, they affect the dimensionality of the input to the first layer. When using LASE E2E, both the GNN and LASE are jointly trained, so the number of learnable parameters are added. All in all, and following the useful notation you introduced, we have that
> > > > - No PE < ASE PE == LASE PE < LASE E2E
> > > > - (L)ASE PE only affects the size of the input in the first layer
> > > >
> > > > Comments along these lines have been included in the re-revised paper; please check the closing paragraph in Appendix A.5.
> > > >
> > > > **Positioning with respect to sub-graph or sub-structure methods**
> > > >
> > > > We appreciate you bringing up the complementary merits of subgraph-based and structural methods, which indeed offer a compelling alternative for inductive settings by focusing on local structural information. We agree that these approaches come with moderate computational cost and address certain limitations of methods that rely on random signal injection. These important points are now reflected in the revised Section 5; please check the paragraphs `On the expressive power of GNNs.’

---

> > > > > ### Author Response · Authors · 2025-05-06
> > > > > **Authors’ response to Reviewer Qvgz’s follow-up comments (part 2 of 2)**
> > > > >
> > > > > **On the use of GAT to explain LASE’s second submodule**
> > > > >
> > > > > This point is well taken. We understand your concern regarding the use of GATs in our framework and appreciate the feedback on the potential for confusion due to the non-standard application of the graph connectivity and attention mechanism.
> > > > > However, we believe that the current framing is crucial to convey the conceptual connection between LASE and existing architectures. While admittedly the attention mechanism is not employed in the standard GAT manner, in our view the GAT reference serves as an analogy to help better contextualize our method within the broader landscape of GNNs. We thus prefer to maintain the current exposition, as we believe this framing best highlights the contribution of our work. Nevertheless, the caption of Figure 3 (the only architectural diagram where the GAT sub-module appears) now explicitly states that it uses a non-standard inner-product-based attention mechanism.
> > > > >
> > > > > Thank you once again for your valuable input.

---

### Review · Reviewer_Rta7 · 2025-02-17

**Summary Of Contributions:**

This paper introduces a novel method, LASE, designed to approximate Adjacency Spectral Embedding (ASE) using a GNN-based approach. ASE aims to generate node embeddings that approximate the adjacency matrix of a given graph. LASE achieves this by integrating GCN and GAT architectures. Its effectiveness and efficiency are empirically evaluated across various datasets, tasks, and optimization techniques.

**Audience:**

Yes

**Broader Impact Concerns:**

I do not have any concerns.

**Claims And Evidence:**

Yes

**Requested Changes:**

Please address each of the concerns outlined in the Weaknesses section above.

**Strengths And Weaknesses:**

**Strengths**
1. The proposed method, LASE, is carefully designed and its architecture and optimization algorithm are clearly explained. This contribution provides valuable insights that can inform further developments in the field.
2. While the primary objective of this paper is to approximate ASE, the authors also demonstrate its application to end-to-end graph representation learning, which adds significant value to practical applications.
3. The paper considers multiple experimental scenarios, including node classification and link prediction, which enhances its overall quality.

**Weaknesses**
1. The motivation for this paper is unclear. While I understand the goal of developing a GNN-based method (LASE) to approximate ASE, it remains questionable why one would opt for LASE over directly performing ASE. The empirical performance differences are marginal (slightly better in some cases but within typical deviation ranges). Clarifying this point would help make the paper more accessible and appealing to the broader ML research community.
2. The discussion regarding the example in Figure 1 is unconvincing. Node clustering on regular graphs without node labels (or node feature vectors) often lacks practical relevance since nodes are topologically identical. As a result, it is unsurprising that effective unsupervised node clustering is not feasible in such cases. The purpose of this discussion should be clarified.
3. One of the target tasks is unsupervised learning, yet there is no sensitivity analysis of parameters of LASE. Such an analysis would be particularly valuable, given the inherent difficulty of parameter tuning in unsupervised settings.
4. The presentation, while generally clear, could be improved. For example, in Section 2.1, the problem is introduced as estimating $\mathbf{X}$ to approximate the adjacency matrix $\mathbf{A}$ of a graph $G$. However, the current phrasing suggests that $\mathbf{X}$ is already given, which is confusing. I had to refer to (Scheinerman & Tucker, 2010) to fully grasp the problem formulation.

---

> ### Author Response · Authors · 2025-04-16
> **Authors’ response to Reviewer Rta7 (part 1 of 2)**
>
> We sincerely appreciate your thoughtful review and constructive feedback. We are pleased that you found our method novel, our contributions valuable in practice and insightful to potentially inform broader impacts to the field, as well as the experimental evaluation comprehensive. We value your suggestions for improvement, and we have implemented revisions accordingly to address the issues raised. Below, we provide responses to your comments and requested changes. We strive to improve our paper and will be happy to continue the discussion if any outstanding issues remain.
>
> **LASE motivation**
>
> This point is well taken and we agree that the motivation behind LASE over the eigendecomposition-based ASE could have been better articulated (especially in the Introduction). We argue that spectral decompositions of the adjacency matrix may face three main limitations:
>
> 1. The eigendecomposition requires full-availability of the adjacency matrix, and it cannot accommodate missing edge data due to e.g., sampling, memory, or privacy constraints.
> 2. High computational complexity, especially in inductive settings whereby one would like to embed multiple moderately large graphs sampled from a common distribution.
> 3. Modifications to the graph require recomputing an eigendecomposition from scratch to obtain the new embeddings (except for specific, e.g., rank-one perturbations).
>
> As we clearly spell out in the revised Introduction, we thus contend that a new architecture is needed when it comes to learning to approximate ASEs. Our objective here is not to propose a new state-of-the-art positional encoding (PE), but rather to broaden the applicability of ASE – whose merits for statistical inference have been well documented.
>
> Limitation 1 is addressed by the low-rank matrix decomposition formulation (1) and the corresponding GD solver in which LASE is based on. By including the mask $\mathbf{M}$, we render LASE applicable to more general graph representation learning settings. Running GD iterations until convergence incurs a computational cost that typically exceeds (highly optimized) eigendecomposition routines that can be used for ASE. However, and as shown in Fig. 8, LASE is an order of magnitude faster than the (more efficient) Block Coordinate Descent variant of GD. Furthermore, the inclusion of attention masks (see Section 3.3) results in shorter computation times than even those optimized eigendecomposition routines in Graspologic.
>
> Finally, since the LASE architecture combines a GCN and a GAT, it may be applied to any graph. Naturally, the accuracy of the obtained embeddings will depend on the underlying distribution. As we empirically show in Section 4.1.2, LASE offers some valuable degree of robustness in this regard. We fruitfully leverage this attractive property to decrease training times (by training on smaller sampled subgraphs of the target large graph to be embedded), thus addressing Limitation 3, as inference under modest changes still yields accurate results. All in all, our experiments provide evidence to support LASE yields node embeddings that are robust to missing data as well as minor distribution shifts, at a fraction of the cost of prior art (addressing Limitation 2) and without sacrificing approximation accuracy.
>
> Following your suggestion, comments along these lines were included in the revised Introduction to better motivate our work. We have also conducted a new experiment, where we study the quality and robustness of the obtained embeddings in the UN dataset, where missing relational data are prominent (and ASE is not directly applicable unless one arbitrarily sets unobserved edges to zero); please check Section 4.1.4. Thanks again for your valuable feedback.
>
> **Clarification of Example 1**
>
> Sorry to hear Example 1 was unconvincing as presented. Regarding regular graphs and the problem of clustering nodes, please note that even if graphs are regular they may exhibit community structure. Example 1 considers a two-class SBM random graph that is regular in expectation. Node classification in this case amounts to determining to which class each node belongs to. A classic method (with theoretical guarantees) is to cluster the nodal representations stemming from ASE. Furthermore, these nodal representations may be used for link prediction as their inner product estimates the edges’ existence probability.
>
> The purpose of Example 1 is to illustrate a setting where a GNN is incapable of obtaining the ASE (i.e., expressing the dominant graph eigenvectors) that is transferable to another graph, even one stemming exactly from the same random graph model. Hence, the need for a modified architecture when it comes to learning to approximate ASEs. The benefits of LASE in this context are discussed in Example 2, which revisits the symmetric SBM setting in Example 1.
>
> Clarifying comments along these lines have been included in the revised Introduction and Example 1.

---

> ### Author Response · Authors · 2025-04-16
> **Authors’ response to Reviewer Rta7 (part 2 of 2)**
>
> **Sensitivity analysis of LASE’s hyperparameters**
>
> The robustness of LASE to its hyperparameters is indeed an interesting aspect worth exploring. Following your suggestion, we have included a new experiment to conduct such sensitivity analysis. In particular, we have trained LASE over a grid of the only two hyperparameters in the architecture: the number of layers $L$ and the embedding dimension $d$. Results for the Cora dataset are reported in Figure 9 of the revised manuscript. We find that – except for limited and somewhat pathological cases (such as excessively large dimension $d$ and small $L$) – the resulting reconstruction loss is fairly insensitive to the choice of hyperparameters.
>
> **Presentation improvements**
>
> Following your suggestion, we have revised the embedding problem statement narrative to make it clear early on that the matrix $\mathbf{X}$ of latent nodal positions is unknown, and the goal is to estimate it from graph observations. Please check the opening paragraph in Section 2.1 of the revised manuscript.
>
> Thanks again for your review.

---

### Review · Reviewer_Dfyx · 2025-04-02

**Summary Of Contributions:**

This paper presents a new algorithm to learn nodal Adjacency Spectral Embeddings (ASE) from graph inputs. The main idea seems to be to leverage Gradient Descent with algorithm unrolling by truncating and re-interpreting
each GD iteration as a layer in a graph neural network (GNN) that is trained to approximate the ASE. The authors term this approach as
Learned ASE (LASE). LASE layers combine Graph Convolutional Network (GCN) and fully-connected Graph Attention Network
(GAT) modules, which is intuitively pleasing since GCN-based local aggregations alone are
insufficient to express the sought graph eigenvectors. Numerical experiments demonstrate
the competitiveness of LASE and its end-to-end variants against other baselines.

**Audience:**

Yes

**Claims And Evidence:**

Yes

**Requested Changes:**

See comments above.

**Strengths And Weaknesses:**

Strengths:
-) This paper is very well-written and easy to follow. Concepts and ideas are communicated well and in a reasonably clear manner. Previous work is cited somewhat adequately.
-) The core of the new algorithm proposed is reasonably novel. Several practical details are discussed.
-) Numerical experiments demonstrate good performance against baselines. Some more effort on comparing against other algorithms, e.g., autoencoder-based, could be beneficial. Also, transformer-based approaches are not really discussed but I do not seem this as a major limitation.

Weaknesses:
-) Section 2 is too long. Nothing new is really being discussed/presented until page 7. Please shorten the section or move some parts to the appendix.
-) Some results obtained via LASE seem to feature high variance (e.g., Figure 10 and elsewhere). Why is that?

General comments:
-) "render eidencomposition of moderately large graphs infeasible". This statement gives the impression that all eigenvectors of the Laplacian must be computed. The real cost of obtaining PEs from graph Laplacian matrices is the cost of applying the Lanczos algorithm for the computation of the eigenvectors associated with a few of the algebraically smallest eigenvalues. This cost is linear to the number of graph edges unless the number of eigenvectors to be computed is too large, in which case the computational cost is dominated by orthogonalization costs.

---

> ### Author Response · Authors · 2025-04-16
> **Authors’ response to Reviewer Dfyx**
>
> Thanks for the time and effort spent in reviewing our manuscript as well as for your thoughtful comments. We are pleased that you found the proposed algorithm novel, our experiments valuable to demonstrate good performance relative to baselines, as well as the paper very well written. We value your suggestions for improvement, and we have implemented revisions accordingly to address all the issues raised. Below, we provide clarifying responses to your comments and requested changes. We strive to improve our paper and will be happy to continue the discussion if any outstanding issues remain.
>
> **Streamlining Section 2**
>
> Your point on the length of Section 2 is well taken. Thanks for the feedback. Following your suggestion, we have implemented edits to streamline the narrative, but admittedly the effective reduction has been minimal (also given that Reviewer Rta7 requested further clarifications on the motivating Example 1). After taking a long, hard look at this background material we believe there is not much more we can cut without sacrificing clarity of exposition leading to (and motivating) the formal problem statement as well as the technical approach in Section 3.
>
> On a related note, we have shortened the Introduction. Specifically, the former opening two paragraphs that focused on the expressive power of GNNs and positional encodings (notions related to, but truly not the main focus of this paper which is to learn to approximate nodal ASEs from graph inputs) were moved to the related work discussion in Section 5. The revised opening paragraph in Section 1 is shorter and more appropriate.
>
> This being said, if you still feel further reductions are needed please let us know in your follow-up comments (specific pointers are much appreciated) and we will be happy to implement the suggested changes to the camera-ready version of the paper.
>
> **High variance in some LASE results**
>
> This is a good observation. Indeed, each datum in the boxplot of the link prediction performance for the UN dataset (Figure 12) includes six randomly sampled countries and 30% of roll calls to predict the corresponding vote. Each such experiment (ten in total) includes different countries (and hence votes), so one expects a non-negligible variability in the results: some countries’ voting patterns are very aligned with allies (in which case the spectral positional encodings should be very informative), whereas others behave in a more isolated manner and their votes will be harder to predict.
>
> Comments along these lines have been included in the revised manuscript; please check the expanded discussion in Section 4.2.1. Holistically and beyond this particular experiment, we cannot see a pattern that suggests LASE intrinsically exhibits significantly higher variability than related approaches (e.g., GD using few iterations or PowerEmbed).
>
> **Complexity of eigendecomposition for large graphs**
>
> Fair point. We agree that the total cost scales linearly with the number of dimensions when the graph is sparse and that not all eigenvectors are necessary to compute the ASE. We have therefore recalibrated the corresponding sentence to explicitly reflect both of these aspects; please check the revised Introduction.
>
>
> Thanks again for your review.

---

> > ### Comment · Reviewer_Dfyx · 2025-05-04
> > **Rebuttal**
> >
> > Thank you for your edits. This is fine with me.

---

### Author Response · Authors · 2025-04-11
**Thanks for the feedback**

We wish to thank the reviewers and the Action Editor for their constructive feedback on the original submission, which contributed to an improved revised manuscript. We are currently finalizing our point-by-point responses addressing each of the reviewers' comments, questions, and requests for changes, as well as revising the paper to incorporate the valuable suggestions provided. Modifications to the revised manuscript will be color-coded blue to ease checking.

We plan to upload our responses and the revised paper incorporating the requested changes in the next few days. We strive to improve our manuscript and will be happy to continue the discussion if any outstanding issues remain.

---

### Decision · Action_Editor_Ptbh · 2025-05-18

**Recommendation:** Accept as is

**Comment:**

The paper was reviewed by three expert reviewers. The reviewers agreed that the paper is easy to read and that the proposed method is clearly explained and is well-designed. They also agreed that the idea behind this paper is promising. However, the reviewers raised concerns about the quality of presentation, the lack of theoretical insights, the lack of experiments on sensitivity, and issues with the empirical validation of the proposed approach. Importantly, some reviewers seemed skeptical about the motivation of this work since it is not clear what are the advantages of the proposed approach over methods that directly compute the spectral embeddings of a graph. Furthermore, one reviewer was concerned about the connection between the layers of the proposed model and the GCN and GAT layers, which actually seems somewhat misleading since the employed attention mechanism is different from the one of the standard GAT layer. Most of these concerns were addressed by the authors in their response, and one reviewer recommended acceptance of the paper, while the other two reviewers recommended weak acceptance. I believe that the submission now meets the TMLR acceptance criteria, and thus, I am recommending acceptance. I personally agree with the reviewers that it should be clearly explained in the manuscript why one should use the proposed model instead of directly computing the eigenvalue decomposition. I also suggest the authors describe in detail how the vanilla GCN and GAT models differ from the corresponding formulations employed in this paper. I also request that the final version of the manuscript carefully considers the reviewers' comments and suggestions.

**Audience:**

Spectral embeddings are very useful in different machine learning applications. Therefore, the findings of this paper will be of interest to some individuals in TMLR's audience.

**Claims And Evidence:**

This paper proposes a neural network model that can be trained to approximate spectral embeddings of graphs. Various refinements (e.g., sparse attention) are also proposed to improve the model's scalability to large graphs. The main claims are supported by clear evidence. The proposed architecture and the optimization algorithm are clearly explained in the submission. The empirical results demonstrate that it can indeed approximate spectral embeddings since its reconstruction error is close to that of the gold standard. It is also empirically shown that the employed refinements lead to faster inference time. For the claims to be even more convincing, I would suggest the authors experiment with other types of graphs beyond SBMs.